

# Accumulation rates over the past 260 years archived in Elbrus ice core, Caucasus

Vladimir Mikhalenko[1], Stanislav Kutuzov[1,2], Pavel Toropov[1,3], Michel Legrand[4,5], Sergey Sokratov[3], Gleb Chernyakov[1], Ivan Lavrentiev[1], Susanne Preunkert[5], Anna Kozachek[6], Mstislav Vorobiev[1],
Aleksandra Khairedinova[1], Vladimir Lipenkov[1,6]

[1]Institute of Geography, Russian Academy of Sciences, Moscow, 119017, Russia
[2]Byrd Polar and Climate Research Center, Columbus, OH 43210, USA
[3]Lomonosov Moscow State University, Moscow, 119991, Russia
[4]Laboratoire Interuniversitaire des Systèmes Atmosphériques, Université de Paris and Univ Paris Est Creteil, CNRS, LISA, F-
75013, France
[5]Institut des Géosciences de l'Environnement, Université Grenoble Alpes, 38402 Grenoble, France
[6]Arctic and Antarctic Research Institute, St. Petersburg, 199397, Russia

*Correspondence to*: Stanislav Kutuzov (kutuzov.1@osu.edu)

**Abstract.** In this study, we present a seasonal-resolution accumulation record spanning the period from 1750 to 2009 Common
Era (CE), based on a 181.8-m ice core obtained from the Elbrus Western Plateau in the Caucasus. We implemented various
methods to account for uncertainties associated with glacier flow, layer thinning, and dating. Additionally, we developed a
novel approach to calculate a seasonal calendar for meteorological data, enabling comparison with ice core records. The
reconstructed accumulation data were compared with available meteorological data, gridded precipitation records, and paleo
reanalysis data. Reconstructed accumulation is representative for a large region south of Eastern European plain and Black sea
region. Summer precipitation was found to be the primary driver of precipitation variability. We identified a statistically
significant but unstable in time relationship between changes in precipitation in the region and fluctuations of the North
Atlantic Oscillation (NAO) index.

## 1 Introduction

Reconstructing past precipitation is a crucial aspect of understanding the Earth's climate system, particularly with respect to
its regional variability. Precipitation plays a critical role in shaping the environment including, water resources, vegetation and
ecosystems. Unlike most key climate indicators which are changing in the same direction in most parts of the world, there are
strong regional variations in the sign of observed changes in precipitation (IPCC, 2014). The spatial distribution of precipitation
is much more variable compared to air temperature and a denser network of observations is required to obtain homogeneous
data. A particularly large mosaic of precipitation records is observed in mountainous areas due to complex interaction of
circulation factors with the underlying surface. Precipitation in situ measurements can be highly uncertain, particularly of
snowfall. This is often due to gauge under-catch, which can lead to significant errors in the measurement of precipitation





amounts (Rasmussen et al., 2012). Currently there are 30 gridded global precipitation data sets are available including gauge-based, satellite-related, and reanalysis data sets (Sun et al., 2018). Despite its discrepancies they are often used for investigating long-term climatic changes. However, their major limitation is generally coarse spatial resolution which is especially crucial

in mountain environments where orographic effects play important role.

Precipitation variability in the pre-instrumental period is mainly obtained from proxy data and climate models (Bunde et al., 2013; Pauling et al., 2006; Valler et al., 2022). Spore-pollen data characterizing the change of plant communities can be used as indirect evidence of the oscillations of wet and dry periods (Barber et al., 2004; Borisova, 2019). They reproduce the qualitative pattern of climatic variability, but generally they have a low temporal resolution (decades, centuries) and do not

always allow quantitative reconstructions.

In contrast, tree rings have an annual resolution and can easily be calibrated using instrumental records. These data are widely used for reconstructions of aridity (Büntgen et al., 2021; Cook et al., 2015, 2020; Solomina et al., 2017) and precipitation (e.g. Touchan et al., 2007; Zhang et al., 2017). The relationship between tree ring parameters of and precipitation strongly depends on the region. Tree-ring width is often a good indirect source of precipitation data in arid conditions, while wood isotope

composition can also be closely related to precipitation in some non-arid regions (Loader et al., 2020).

Unlike other proxy, glaciers contain a direct precipitation signal. Annual layer thickness in ice cores depends on total precipitation amount although annual precipitation not always equal to net accumulation. The most accurate data can be obtained in areas where the snow mass loss due to melting, sublimation, wind and avalanche snow redistribution is minimal. Many polar ice cores were recovered in the central parts of the ice sheets near the ice divides where there is a balance between

snow wind erosion and accumulation. On mountain glaciers, the underestimation of wind-driven snow erosion can lead to significant errors.

To obtain past accumulation rates, the annual-layer thickness has to be corrected for the cumulative effect of ice flow. The algorithm for calculating the initial thickness of deposited annual layers at the surface is quite well developed (Dansgaard and Johnsen, 1969; Nye, 1963; Paterson and Waddington, 1984; Schwerzmann et al., 2006). The accuracy of accumulation

reconstruction depends on the use of ice flow models to estimate the displacement of the drilling site due to the movement of the glacier and the thinning of the annual horizons, especially for the deep parts of the glacier (Licciulli et al., 2019). For these reasons detailed ice-core reconstructions of accumulation and precipitation are relatively rare (Dahl-Jensen et al., 1993; Goodwin et al., 2016; Henderson et al., 2006; Pohjola et al., 2002; Winstrup et al., 2019; Yao et al., 2008) compare to other climate and environmental parameters.

In this paper, we present a high-resolution reconstruction of snow accumulation based on the ice-core records from Mt. Elbrus, Caucasus and interpretation of the obtained results.





## 2 Data and Methods

### 2.1 Site description

The Caucasus mountain system is situated between the Black and the Caspian seas, and generally trend east-southeast, with

the Greater Caucasus range often considered as the divide between Europe and Asia. The 2223 glaciers in the Caucasus cover an area of $1060.9 \pm 33.6$ km$^2$ (Tielidze et al., 2022). The Elbrus mountain glaciers contain about 10% of the Caucasus ice volume and cover an area of 112.6 km$^2$ (Mikhalenko, 2020) (Fig. 1a). Glaciers cover altitudinal range from 2683 to 5642 m asl with the coldest conditions present above 5200 m asl where mean summer air temperature stays below 0°C. The ice cores used for climate and environment reconstructions were recovered at two sites: Western Plateau (WP) at 5115 m asl and the

glacier in the crater of the eastern summit of Elbrus at 5600 m asl (Mikhalenko et al., 2021). The WP area is ~ 0.5 km$^2$ and is surrounded by lava ridges to the south and southeast, and by a vertical wall of Mt. Elbrus to the east (Fig. 1b). Ice thickness according to ground-based survey in 2004-2007 varies from 100 to 200 m with a maximum of $255 \pm 8$ m in the north-eastern part of the plateau (Lavrentiev et al., 2010). 10-m depth temperature is –17.3°C (Mikhalenko et al., 2015). A 181.8 m ice core was recovered at the WP in August-September 2009 and the crater in August 2020 (Mikhalenko et al., 2020).

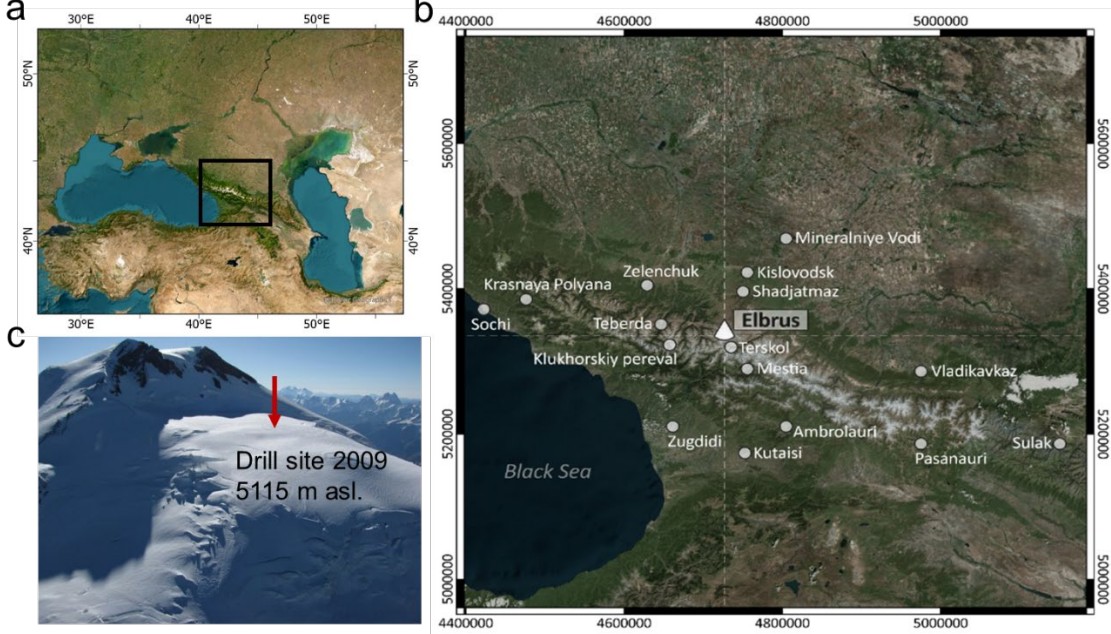


**Figure 1: Location of the Mt Elbrus drilling site: (a) the Caucasus; (b) glaciers and meteorological stations; (c) Western Elbrus plateau drill site (photos by I. Lavrentiev, September 2009). ArcGIS World Imagery Basemap used as the background. Source: DigitalGlobe.**

The amount of precipitation can be determined as the difference between the measured accumulation layer and the loss caused

by sublimation, evaporation, and wind-driven snow redistribution. On the WP in 2018, water evaporation during the short season of possible surface melting was estimated to be 3% of the accumulation layer or 45 mm w.e a$^{-1}$. Sublimation of ice





crystals during snowstorms was measured as 100 mm w.e. or 7% of the accumulation (Mikhalenko, 2020). The contribution of sublimation rate to glacier mass balance and snow cover balance is estimated to be between 5-10% (Bintanja, 1998; Palm et al., 2017) but can reach up to 30% in certain climatic conditions (Pomeroy and Gray, 1995).

Wind-driven snow redistribution was measured by stakes on the plateau during three field seasons, showing a zero balance between scouring and accumulation of snow near the drilling site (Mikhalenko, 2020). An analysis of the fields of summer and winter accumulation on the plateau from 2015-2017 shows that snow accumulation in the summer period is higher than in winter and occurs more evenly over the plateau, with its distribution being more stable from year to year. In winter, the maximum snow accumulation shows a clear shift to the northern and eastern parts of the plateau, where it is limited by the

northern ridge and the steep wall of the Western summit of Elbrus. In the southern and western parts of the plateau, absolute minima are observed in the winter accumulation fields, which are likely caused by strong winds during winter (Lavrentiev et al., 2022). The area near the drilling site is characterized by mean values of snow accumulation. The total value of snow loss on the WP is estimated to be about 10% (Mikhalenko, 2020). Although we cannot rule out the higher snow accumulation losses for winter layers at the drilling site.

## 2.2 Ice core analysis


The 2009 WP ice core was processed and analysed at the Institute of Environmental Geosciences in Grenoble, France for major ions ($K^+$, $Na^+$, $Ca^{2+}$, $Mg^{2+}$, $NH_4^+$, $SO_4^{2-}$, $NO_3^-$, $Cl^-$, $F^-$), succinic acid ($HOOCCH_2COOH$), dust concentration, and black carbon (Kutuzov et al., 2019; Lim et al., 2017; Preunkert et al., 2019). Stable isotopes ($\delta^{18}O$ and $\delta D$) were analysed using Piccarro at the Arctic and Antarctic Research Institute in St. Petersburg, Russia (Kozachek et al., 2017). Determination of the

tritium content was carried out at the University of Bern, Switzerland (Mikhalenko et al., 2015). 3724 samples were analysed in total down to the 168.6 m depth. Sampling resolution was 10 cm at the upper part of the ice core and decreased to 5 cm at 70 m depth and to 2 cm at 157 depth and below. Discrete sampling in the cold room was performed according to clean sampling protocol (Preunkert and Legrand, 2013). Detailed results of ice-core chemical records are given in (Preunkert et al., 2019).

### 2.2.1 Ice core dating

The upper 168.55 m (131.56 mwe) depth of the ice core were first dated by annual layer counting primarily using pronounced seasonal variations in ammonium and succinate concentrations, both exhibiting well-marked winter minima (Mikhalenko et al., 2015; Preunkert et al., 2019). Ammonium can reach the WP (5115 m) as a result of deep convection or with streams of well-developed mountain-valley circulation. Both phenomena are observed in most cases in the warm half of the year and their intra-annual frequency and intensity are closely related to the seasonality of free convection in the atmosphere (Fagerli et al.,

2007). The very low winter $NH_4^+$ levels are related to precipitation of the cold half-year.

The annual counting was found to be accurate (a 1-year uncertainty) over the last hundred years when anchored with the stratigraphy of the 1912 CE Katmai horizon located at 116.72 m (87.75 mwe) depth (Mikhalenko et al., 2015). Although several other volcanic horizons were suspected to be recorded, particularly below 137 m (105 mwe) depth, none of them was





unambiguously attributed to a particular event. A series of sulfate spikes present between 152.58 and 154.92 m (118-120 mwe)
depth dated at ~ 1840-1833 CE (Table S1), of which two were characterized by an increase acidity (up to 7.8 µEq L$^{-1}$), were, however, suspected to be possibly related to the 1835 CE Coseguina eruption. It has to be emphasized that given the time period covered by the ice core (1774-2009 CE). Surprisingly the layer corresponding to the large eruption of Tambora (1815 CE) was not identified in previous dating of the Elbrus ice core. Uncertainty was not estimated for the dating below the 1912 CE Katmai horizon.

With the aim to estimate uncertainties in dating established from annual layer counting prior to 1912 CE, we revisited the chemical profiles including sulfate and acidity. By its nature, dating using annual layer counting becomes more uncertain with depth because identification of winter layers is less straightforward due to the decrease of annual layer thicknesses resulting from glacier ice flow (e.g. Paterson and Waddington, 1984). To evaluate the subjective character of the annual layer counting method, four co-authors (denoted A1, A2, A3, and A4) performed the annual layer counting using seasonal variation of
ammonium and succinate concentrations without considering a strict criterion. The results were compared to those (denoted A5) previously proposed by Preunkert et al. (2019). As seen in Figure 2, the five chronologies are quite consistent until 131.5 m (100 mwe) depth dated at 1890 CE ± 3 years. The dating uncertainty increases with depth, the layer at 143.48 m (110 mwe) and 154.93 m (120 mwe) depth being dated at 1867 CE ± 6 years and 1828 CE ± 15 years, respectively. A3 and A4 independently counted more years than A1, A2, and A5 by attributing much more ammonium and succinate minima to winter
layers (S1). At 168.55 m (131.56 mwe), A1, A2, and A5 proposed an age of 1774 CE ± 6 years, whereas A3 and A4 proposed an age of 1750 CE ± 4 years (i.e., a departure of ~25 years).

Using this new dating we attributed layer of increased acidity and sulfate located at 112.57 mwe (dated at 1851 CE) to the 1854 CE Shiveluch eruption (Kamchatka) of which the volcanic exclusivity index (VEI) was of 5 (Simkin and Siebert, 1994). The series of spikes located between 118 and 120 mwe was previously identified as possibly related to the 1835 CE Cosigüina
eruption are now attributed to the 1815 Tambora eruption. The layer dated at 1789 CE may be related to the 1783 CE Laki eruption (see further discussions in supplementary).

The WP chronologies are consistent within one or two years back to the horizon of Shiveluch in 1854. Further down, dating differ by 25 years at 130.9 mwe (1750 CE ± 4 years instead of 1774 CE ± 6 years), depending on the results of the annual layer counting. The new dating (1750 CE ± 4 years) is consistent with a Tambora layer located at 118.96 or 119.84 mwe and possibly
a Laki layer at 124.71 mwe. Use of the two possible dating only slightly affects the general trends and tendencies of the accumulation reconstruction (S4). Further in the paper we use new dating as a base for the accumulation reconstruction.





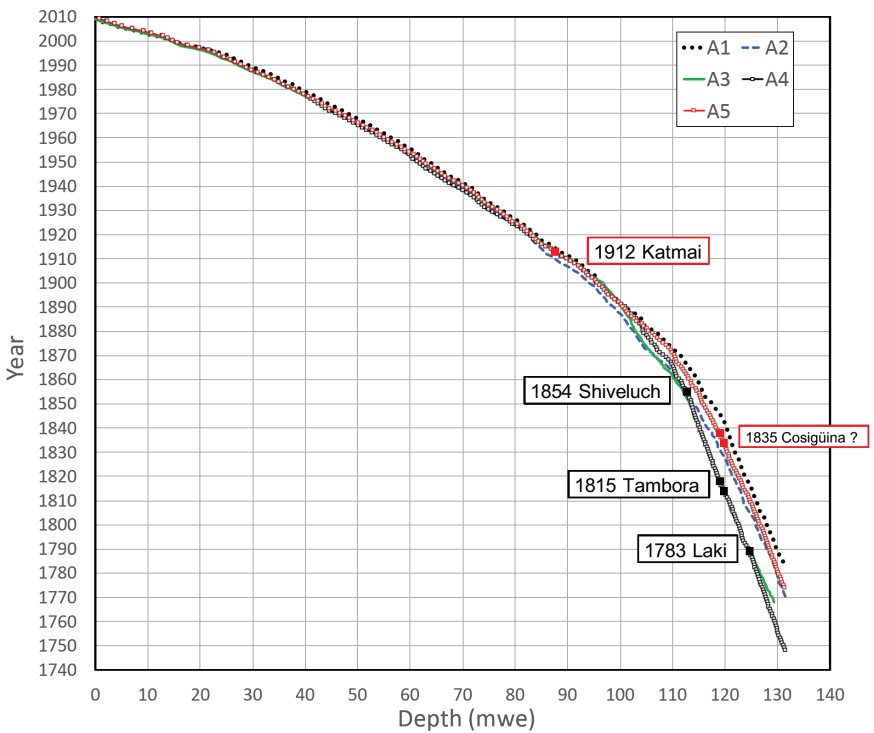

Figure 2. Age-depth (in mwe) relation of the Elbrus ice core derived by annual layer counting achieved by 4 co-authors of the present paper (denoted A1, A2, A3, and A4) and by Preunkert et al. (2019) (denoted A5).

**2.3 Snow accumulation reconstruction**

**2.3.1 Thinning of annual layers due to ice flow**

The accumulation rate history at Mt. Elbrus can be inferred from depth profiles of annual-layer thicknesses in the WP ice core when corrected for firn densification and thinning of layers due to ice flow. Density measurements, as reported by Mikhalenko et al. (2015), reveal kinks in the curves of the studied ice cores, corresponding to critical density values of 550 and 840 kg m$^{-3}$ (Maeno and Ebinuma, 1983). The third critical density value of 730 kg m$^{-3}$, which marks the transition of firn into ice by complete closure of air inclusions, is not clearly visible on the density curve. This indicates ice formation without meltwater involvement (Hörhold et al., 2011; Ligtenberg et al., 2011). The depth at which air bubbles separate from the surrounding ice matrix and pore closure occurs is approximately 55 m, with a measured density of about 840 kg m$^{-3}$ (Mikhalenko et al., 2015). Although the annual layer thickness exhibits high variability, the data suggest that layer thinning occurs with increasing depth due to ice flow. To determine snow accumulation values, we used a simple J. Nye flow model (Dansgaard and Johnsen, 1969) incorporating mean accumulation rate and constant glacier thickness over the time span represented by an ice core:

$$h_r = h_o \times e^{\frac{h_o \times A}{H}}, \tag{1}$$

$$h_o = \frac{H \times W\left(\frac{A \times h_r}{H}\right)}{A}, \tag{2}$$



where $h_r$ and $h_o$ – recent and original depths of annual layer respectively (m), $A$ – age (years), $H$ – depth of glacier, $W$ – Lambert W function.

Scaling of the best fit curve and the residuals of the raw data from the best fit curve (S3a) to the linear mean accumulation

value was considered as the representation of the variability of the "original" annual layers' depths as if no variability in accumulation over the glacier surface was expected (S3b).

### 2.3.2 Calculation of backward trajectories

Since the ice core includes layers that were deposited upstream of the drilling site, where annual snow accumulation conditions may differ from those at the drilling site, it is crucial to identify the ice core catchment area in order to investigate the connection

between the ice core data and surface accumulation. To verify the representativeness of the ice core for the accumulation conditions at the drilling site and account for upstream effect, we reconstructed the backward trajectories of the ice flow. To this end, we calculated the backward trajectories of the ice/firn particles positioned along a vertical line segment connecting the drill site and the bedrock (Fig. 3). In the post-processing step, the backward trajectories were calculated by utilizing the Runge-Kutta integrator implemented in the stream tracer of the ParaView visualization application, based on the modeled

glacier velocity field. (Ahrens et al., 2005). The velocity field is simulated based on a 3-D full Stokes ice-flow model with the firn rheological law (Gagliardini and Meyssonnier, 1997). The model is implemented using the finite element software Elmer/Ice (Gagliardini et al., 2013). We performed a steady-state simulation with fixed glacier geometry.

The mathematical formulation of the ice-flow problem follows (Zwinger et al., 2007) with some simplifications and includes the Stokes and the volume balance evolution equations, the stress-free surface boundary condition and the no-slip bedrock

boundary condition. Unlike (Zwinger et al., 2007), we used only dynamical equations and did not consider thermo-mechanical coupling. The model used an approximation of the density profile measured in the borehole drilled in 2009 (Mikhalenko et al., 2015). Model simulations show that constant value of the flow rate factor in the firn rheological law has very little effect on the geometry of the backward trajectories and the position of their sources on the glacier surface. Therefore, the rate factor may be chosen arbitrary from a wide range of values.

Since the digital elevation model data (Lavrentiev et al., 2022) covered only part of the glacier bedrock, the computational domain was restricted to this area, and the lateral boundary connecting the surface and the bedrock boundaries was a fragment of a cylindrical surface (vertical wall) (Fig. 3c). As most of the lateral boundary was not a physical boundary in the glacier, the flow of ice/firn through it was possible. We considered two alternative boundary conditions for the lateral area: (i) no outflow (zero horizontal velocity) at the whole lateral boundary, and (ii) constant nonzero outflow velocity at the western and

southern parts of the lateral boundary and no outflow at the eastern side where a vertical wall of Mt. Elbrus is situated (Fig. 3a). In case (ii), the outflow velocity vectors were directed outward from the computational domain along the normal to its lateral surface. The outflow velocity value was considered constant in space and time.



Our simulation showed that the geometry and the length of the backward trajectories were influenced by the presence or absence of the outflow rather than the particular value of nonzero outflow velocity. Therefore, this velocity value was not an essential parameter in our modeling. Additionally, no field data on velocities were available.

In our model experiments, we obtained the backward trajectories of maximal length under conditions (ii). Since the maximum extent of the ice core catchment area was of interest in our study, we present the results of this model case. We used the flow rate factor $B = 4.82$ MPa$^{-3}$a$^{-1}$ and the outflow velocity $v_{out} = 1$ ma$^{-1}$ in the case illustrated in Figure 3.

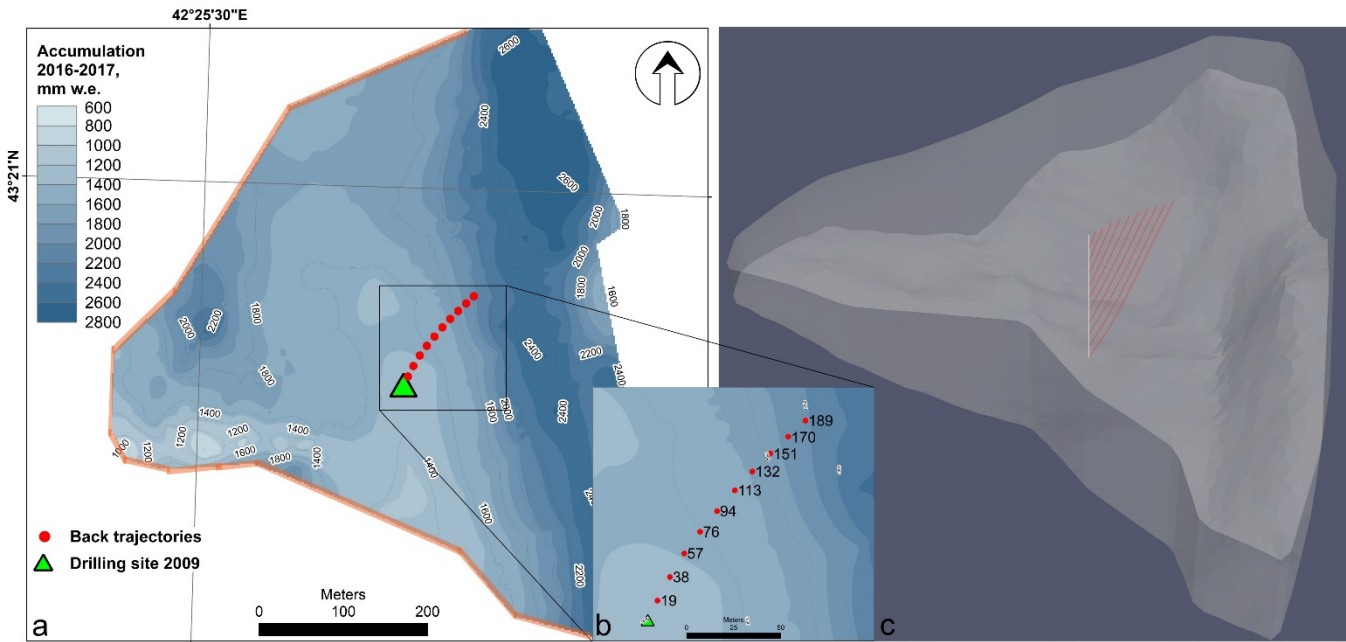

**Figure 3. Accumulation map for the Western plateau of Elbrus in 2016–2017 CE (Lavrentiev et al., 2022) (a); the modeled catchment area for the 2009 CE ice core (b); and the computational domain (c) with the borehole position (white line) and the calculated backward trajectories (red lines). An orange line shows the outflow zone. The source points of the backward trajectories are marked with red dots; the corresponding borehole depths (rounded, in m) according to DEM (2017) are given as dots labels. The green triangle indicates the position of the ice core (2009).**

Based on our simulation, we have determined that the sources of the backward trajectories are located on the glacier surface, no more than 140 meters northeast from the drill site used in 2009 CE. To estimate the correction factor for the difference in snow accumulation at the initial source location (S5), we utilized average accumulation distribution maps, which were generated using a combination of ice core data and a high-frequency radar survey (Lavrentiev et al. in 2022). Snow accumulation increases linearly with distance from the drill site along the backward trajectory. Therefore, we applied the same linear relationship to calculate the accumulation correction (%) based on the ice age. The most significant correction of 30% was applied to the oldest layers dating back to 1750 CE (S6).



## 2.4 Climate data

215 Data from 16 weather stations located in the foothill or mountainous regions of the Caucasus (Russia and Georgia) were used for the statistical estimates of the precipitation field in the North Caucasus, as well as for comparing the time series of precipitation with snow accumulation on the WP drilling site (Table 1). Most of the stations are located in lowland and foothill areas (less than 1000 m asl), 5 - in the middle mountain zone, and 4 - in high mountains (more than 2000 m asl). The CAPE (convective available potential energy) was calculated for separating summer and winter seasons using the ERA5 reanalysis

220 data. We also used Global Precipitation Climatology Centre (GPCC) v2020, 0.25° monthly precipitation dataset from 1891-present calculated from global station data for field correlation analysis.

**Table 1. Meteorological data used for analysis**

| Weather station or reanalysis | Distance from drilling site, km | H, m asl | Coordinates | Observation period | Values |
|---|---|---|---|---|---|
| Terskol (RUS) | 12 | 2350 | 42.73N; 42.33E | 1951–2010 | P |
| Klukhorskiy pereval (RUS) | 50 | 1815 | 43.15N; 41.52E | 1965–2010 | P |
| Sulak (RUS) | 250 | 2927 | 42.22N; 46.25E | 1936–2010 | P |
| Teberda (RUS) | 60 | 1325 | 43.27N; 41.25E | 1926–2010 | P |
| Shadjatmaz (RUS) | 45 | 2056 | 43.7N; 42.40E | 1961–2010 | P |
| Kislovodsk (RUS) | 75 | 860 | 43.90N; 42.50E | 1936–2010 | P |
| Mineralnie Vody (RUS) | 100 | 420 | 44.10N; 43.10E | 1955–2010 | P |
| Krasnaya polyana (RUS) | 170 | 960 | 43.60N; 40.20E | 1936–2010 | P |
| Sochi (RUS) | 190 | 50 | 43.50N; 39.60E | 1875–2010 | P |
| Zelenchukskaya (RUS) | 70 | 925 | 43.90N; 41.50E | 1960–2010 | P |
| Vladikavkaz (RUS) | 170 | 750 | 43.00N; 44.60E | 1951–2010 | P |
| Mestia (GEO) | 40 | 1445 | 43.00N; 42.60E | 1961–2010 | P |
| Kutaisi (GEO) | 105 | 260 | 43.20N; 42.60E | 1936–2010 | P |
| Zugdidi (GEO) | 125 | 160 | 42.50N; 41.90E | 1961–2010 | P |
| Ambolauri (GEO) | 85 | 544 | 42.50N; 43.00E | 1936–2010 | P |
| Pasanauri (GEO) | 130 | 1070 | 42.40N; 44.60E | 1936–2010 | P |
| Reanalysis CFSR | 30 [*] | | | 1979–2010 | T, E, f, q [**] |
| Reanalysis ERA5 | 15 [**] | | | 1979–2010 | CAPE [***] |
| GPCC | | | | 1891-2010 | P |

225 [*] The distance between the drilling site and the nearest node of the reanalysis grid.





** T - temperature, E - vapor saturation pressure, f - relative humidity, q - cloud air mixture ratio
*** CAPE - convective available potential energy
RUS – Russia
GEO - Georgia

### 2.4.1 Seasonal calendar

A direct comparison of the ice-core reconstructed net accumulation/precipitation records with precipitation amounts measured at meteorological stations contains seasonal and annual dating uncertainty. The boundaries of warm and cold seasons at meteorological stations can vary significantly depending on the geographical location and altitude. Seasonal variations of ammonium for which the main source is vegetation in the active phenological phase corresponding to the warm half-year (April-September) were used for identification winter and summer layers in WP ice core. Ammonium concentration in the ice core is maximum during the period of active convection and the accumulation sum over this period corresponds to precipitation of the warm half of the year.

As a seasonality predictor, we used the convective available potential energy (CAPE) calculated from the ERA-Enterim reanalysis data for 1979–2010 which has a pronounced seasonal variation and reflects convective movements in the atmosphere due to buoyancy forces. CAPE depends on the difference between the virtual temperature of the air particle ($Ti$) and the virtual temperature of the air surrounding the particle ($T$). Integrating this difference along the vertical ($z$) and multiplying it by the gravitational acceleration $g$, we obtain the convective instability energy in $Jkg^{-1}$.

$$\int_{z1}^{z2} g\left(\frac{Ti-T}{T}\right)dz \tag{3}$$

A number of studies (Chen et al., 2008; Markowski and Richardson, 2010)formulate gradations of CAPE that correspond to a particular level of convective atmospheric instability. Most of these studies are aimed at identifying the relationship between the recurrence of hazardous weather events of convective origin and the CAPE value. For example, in (Chen et al., 2020) it is shown that most of the weather hazards in the temperate US climate zone occur when the CAPE is between 260-840 $Jkg^{-1}$. Under these conditions there is a high probability of deep convection development that can transport biogenic particles and chemical ions from the ground layer of the atmosphere to the 5000 m level. At Caucasus latitudes the CAPE may exceed the value of 300 $Jkg^{-1}$ even in the cold half of the year, on some exceptional days but not systematically. We therefore assume that a warm half-year in the North Caucasus begins when the CAPE exceeds 300 $Jkg^{-1}$ at least once every five days, i.e., during the natural synoptic period. Analysis of the annual variations of CAPE over the Caucasus (S7) shows CAPE is consistently less than 300 $Jkg^{-1}$ in summer in the region only under long and pronounced anticyclonic conditions corresponding to blocking. Blocking anticyclones are characteristic of more northern latitudes, with blocking developing much less frequently in the subtropics and in the southern temperate climate zone (Bardin et al., 2019). In addition to deep convection, the advection of ammonium ions to the WP level may be regulated by the mountain-valley circulation, which is also most active in the summer months and is indirectly related to the CAPE value



The CAPE value depends on the heat content of the lower atmospheric layers and the intensity of heating of the underlying surface, so it has a pronounced seasonal variation. In spite of significant climate changes in the Caucasus, the CAPE value, which is one of the thermal characteristics of the atmosphere, did not show statistically significant increase neither in the duration of the warm period (number of days) nor in the seasonal mean value. For the period covered by the reanalysis data (1979-2015), a calendar of warm and cold seasons was compiled (S8).

## 3 Results and discussion

### 3.1 Net accumulation reconstruction

The graphical results of the accumulation reconstruction are presented in Figure 4. The dating accuracy of the reconstruction varies, with a precision of ±1 year for the years 2009-1912 CE, ±2 years in the period 1912-1825 CE, and a decrease to ±4 years in the period 1825-1750 CE. The mean annual accumulation reconstructed at the WP from 1750 to 2009 CE is 1.641

m.w.e. During this same period, the mean summer and winter accumulations are 1.156 m.w.e and 0.485 m.w.e, respectively. Due to dating uncertainties this record is suitable for investigations of decadal, multidecadal and long term regional precipitation variations rather than interpretations of the accumulation for the exact years.

No statistically, significant trends were observed in 20th century in WP. A slight positive trend (0.018 m w.e. per decade) was estimated for the annual accumulation over the record, attributable to a general increase in winter accumulation. However, this

trend is likely due to the insufficient sampling resolution of the deepest layers, which failed to fully capture the winter layers for a certain year and resulted in an underestimation of winter accumulation below 110 mwe depth (corresponding to the year 1865 CE).

The record can be classified into five major periods based on accumulation amount and year-to-year variability. The period before 1830 was characterized by increased summer accumulation and annual variability. This was followed by the relatively

low summer and winter accumulation of the 1830-1860 CE period with less pronounced variations. From 1860 to 1935 CE, summer and winter accumulation increased by a factor of two and strongly fluctuated. The period from 1935 to 1980 CE was characterized by low accumulation in summer and relatively high accumulation in winter. Since 1980 CE, an increase in summer and winter accumulation has been observed.

The WP site exhibits a seasonal distribution of precipitation typical of the Central Caucasus, with convective precipitation

leading to a maximum in the summer months. The mean share of summer accumulation in the total annual accumulation was 70% (STD=18) over the course of 260 years, which is consistent with current measured precipitation data at weather stations. Over the entire period covered by the core data, there is a statistically significant decrease in the contribution of summer accumulation to the annual total ($R^2 = 0.6$). The highest percentage of summer accumulation occurred between 1750 and 1830 CE, with an average contribution of 83% to the annual total, resulting in an anomaly of over 200% compared to the modern

level in some years. In contrast, the lowest percentage of the summer component of the annual accumulation was observed in 1935-1980 (57%). This period was characterized by a prolonged negative anomaly of summer accumulation (Fig. 4b), the



average value of which was -25%, surpassing the natural interannual variability of ±20% for the Central Caucasus (Toropov et al., 2019).

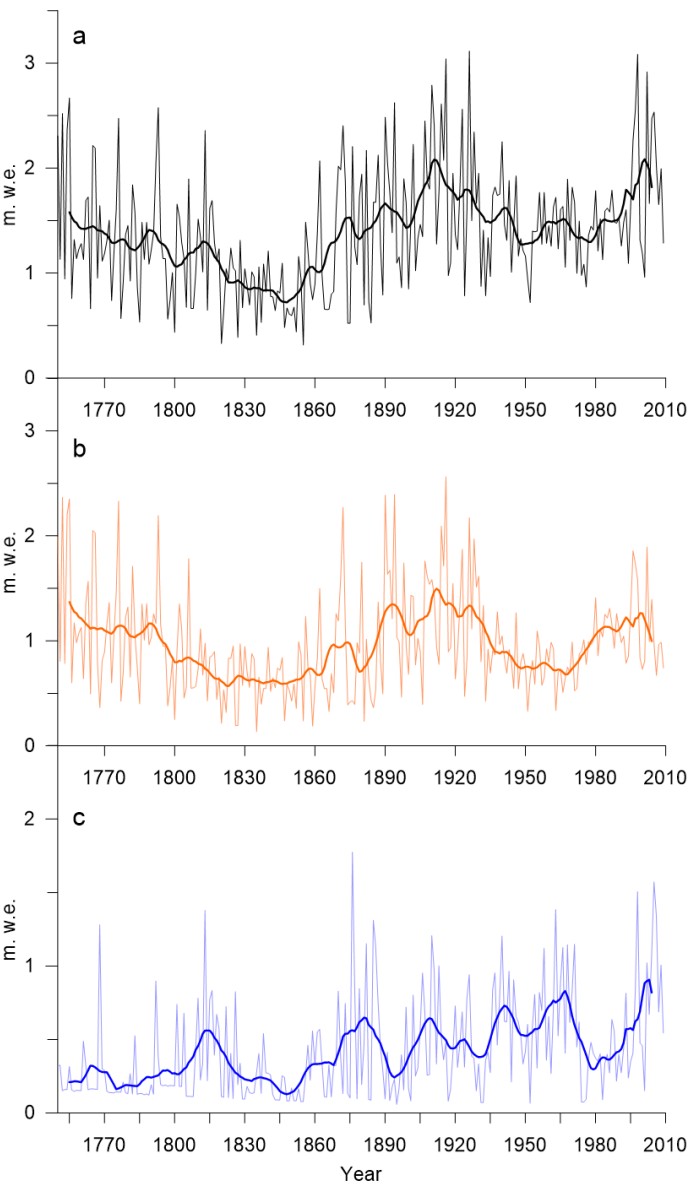

**Figure. 4. Reconstructed annual (a), summer (b) and winter (c) snow accumulation at the WP. 10-year moving averages are shown in thick lines.**



### 3.2 Comparison with meteorological data

#### 3.2.1 Spatial distribution of precipitation in the North Caucasus

The interpretation of the reconstruction of accumulation relies on a comparison with meteorological information gathered
during the period of instrumental observations, utilizing weather station data, global precipitation data sets and reanalyses. This comparison enables us to assess the accuracy of annual and seasonal accumulation amounts, as well as to determine the area where this reconstruction holds significance for different timescales. Additionally, we can establish the statistical and physical relationship between resulting accumulation and precipitation amounts, while mitigating the potential impact of confounding factors such as wind drift, avalanche supply, evaporation, and sublimation rates.

The spatial correlation function for the observed annual precipitation amount was calculated relative to the Terskol meteorological station closest to the drilling site. The spatial distribution of the normalized correlation coefficients between the series of annual precipitation in Terskol and at other stations in the Caucasus is shown in Fig. 5.

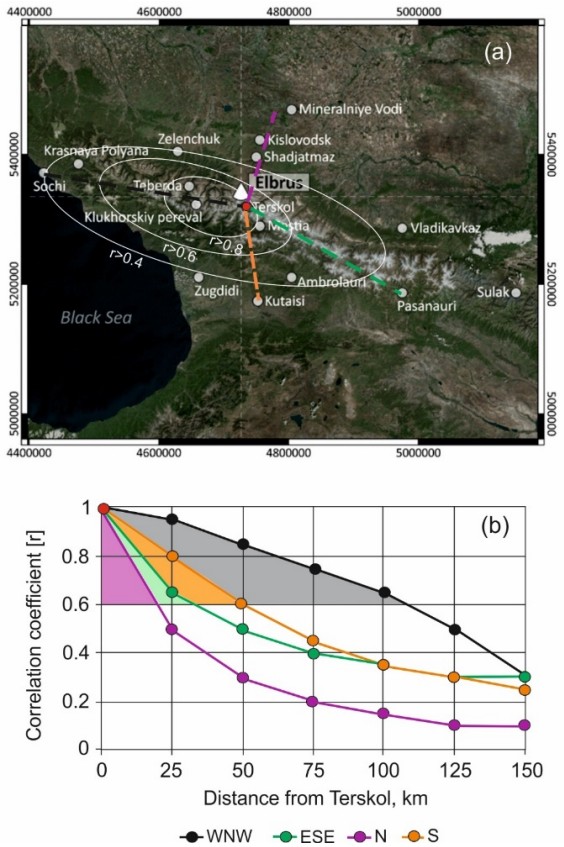

**Figure. 5. Spatial correlation function of the annual precipitation calculated relative to the Terskol meteorological station (red dot)**
**(a): isolines of correlation coefficients calculated according to Student's t-test are given by white lines; dashed lines indicate the profiles for which the correlation function was calculated. Correlation function values along these profiles (b): filled areas correspond to areas with statistically and physically significant correlation (r > 0.6). ArcGIS World Imagery Basemap used as the background. Source: DigitalGlobe.**



The significance of the normalized correlation coefficient was assessed using Student's t-test, with a threshold value of 0.4.

However, as a correlation coefficient exceeding this value not only has statistical but also physical meaning, the threshold was increased to 0.6. Figure 5 demonstrates that the precipitation field in the Caucasus is significantly isotropic, with correlation coefficient isolines appearing as elliptical shapes stretching along the Greater Caucasus. The correlation radius is larger in the WNW direction, owing to the influence of western air mass transport and the Black Sea, compared to the ESE, S, and N directions, as moisture-bearing air masses rarely reach the Greater Caucasus from the north. Consequently, annual year-to-

year variability of precipitation in the northern foothills may differ from the precipitation in the high-mountain regions of the Caucasus. Despite the challenging orographic conditions, the radius of statistically significant correlation of annual precipitation ($r > 0.4$) measured at the weather stations encompasses the majority of the Western and Central Caucasus, while a physically significant radius ($r > 0.6$) extends to 50-100 km. Similar findings were reported by (Tashilova et al., 2019). The highest correlation ($r \geq 0.8$) occurs within an area of 10-50 km, indicating that on the annual and sub-annual timescale ice-core

records likely provide description of precipitation within 50-100 km radius.

Statistical analysis of long-term series of annual precipitation (Sochi, Teberda, Krasnaya Polyana) and reconstructed accumulation on WP reveal high similarity of empirical distribution functions (Fig. 6), which are asymmetric log-normal distributions (Fig. 6 b, c) shifted towards positive anomalies and decreasing sharply towards negative ones. The median value corresponds to a positive anomaly of 10-15%, which is typical for time series of precipitation. The root mean square (RMS)

deviation of both precipitation and reconstructed accumulation is approximately 20%, which is typical for annual precipitation in temperate climatic zones. The range of variability of precipitation anomalies, based on averaged weather station data, ranges from a decrease of 50% to an increase of 80%. In contrast, ice-core records show a significantly larger range of variability, from a decrease of 80% to an increase of 200%. This difference can be attributed to the length of the ice-core data, which spans 260 years, significantly longer than the longest continuous meteorological records. Consequently, the ice-core data captures

the full range of precipitation variability in the Caucasus, ranging from extremely dry conditions (one-fifth of annual precipitation) to exceptionally wet conditions (double the annual average).





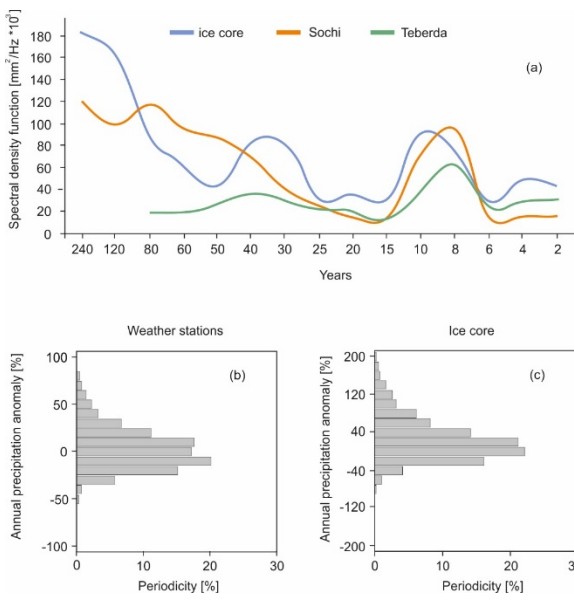

**Figure 6. Statistical analysis of the data time series on the longest-range weather stations and ice cores: (a) spectral analysis of the series, (b) empirical distribution function of annual precipitation anomalies compared to the period 1981-2010, averaged between data from the longest-row weather stations Sochi, Krasnaya Polyana, Teberda and Sulak-Vysokogornaya, (c) empirical distribution function of annual precipitation anomaly on the West Elbrus plateau according to ice core data compared to the period 1981-2010**

Spectral analysis showed that according to the accumulation data and the results of meteorological observations, the statistically significant peak of the spectral density function for all data series (ice core, Sochi, Teberda) falls on a period of 8-10 years, and 30-40 years. The second period is close to the characteristic scale of the North Atlantic Oscillation (NAO) and Atlantic Multi-decadal Oscillation (AMO) index fluctuations. The similarity of the distribution functions and spectral density from the data of weather stations and ice-corer records indicates that the reconstructed accumulation values at Elbrus are primarily related to precipitation rather than to other accumulation factors. In some years, they may play a decisive role, but at averaging scales of more than 5 years, the precipitation factor plays the predominant role.

### 3.2.2 Seasonal accumulation

Over the past 47 years, the period of summer identified by CAPE typically spans from May 1 to October 10, which nearly aligns with the hydrological year. However, our calendar displays considerable inter annual variability. The earliest sustained transition through 300 Jkg$^{-1}$ occurred on April 1, 2008, while the latest was on May 27, 2003. The earliest transition to winter convective instability took place on September 17, 1986, and the latest on November 1, 2005. The minimum period with steady CAPE > 300 Jkg$^{-1}$ was 135 days (in 2010), the maximum was 200 days (in 1995 and 2005), and the average was 164 days. The distribution of the warm period's duration by CAPE is close to normal, but at the 2σ level, the deviation is ±1 month. Therefore, accounting for this factor in determining seasonal and annual precipitation amounts is critical. Failure to do so can result in errors exceeding 10% of annual precipitation sums, as illustrated in Figure 7. This figure displays regression relations





of the accumulation reconstructed from the core with precipitation amounts at the nearest weather stations for the 1979-2009

CE period.

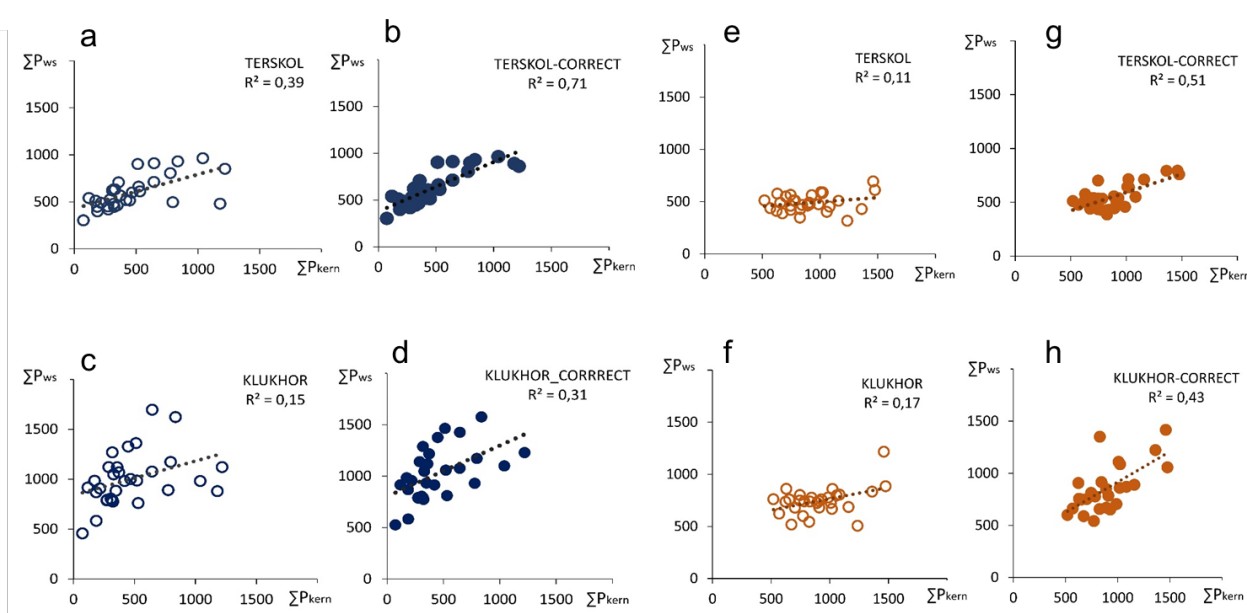

**Figure. 7. Regression relations between the accumulation layer on the West Elbrus plateau (∑Pkern, mm) and precipitation amounts at the nearest weather stations (∑PWS, mm): (a-d) for the cold season (a, c for October-March calendar period, b, d for the cold period highlighted by CAPE value thresholds); (e-h) for the warm season (e, f for the April-September calendar period, (g, h) for the warm period highlighted by CAPE values).**

It is well seen that the $R^2$ in the relations between precipitation and accumulation in the ice core with considering CAPE dates increases, and become statistically significant (more than 0.4, p<0.001). This indicates that accounting for the role of free

convection (CAPE) significantly improves the relationship between accumulation on the WP and precipitation measured at weather stations. The improvement is most significant for the winter season. This is due to the greater spatial homogeneity of the precipitation field in the cold half of the year, which is expressed in larger values of the radii of significant correlation. In summer likely due to a significant contribution of local convective precipitation the improvement is not so significant. To facilitate further analysis, we have defined the cold period as the period from October to March and the warm period as the

period from April to September.

**3.2.3 Comparison with gridded data**

Spatial comparison with the gridded GPCC v2020 0.25 precipitation dataset revealed a statistically significant correlation for both seasons between ice core accumulation and precipitation estimates (Fig. 8). Summer accumulation moderately correlates with precipitation observations over Ukraine, South Russia, and the Black Sea. Winter accumulation shows an even stronger

correlation with gridded datasets over the entire Black Sea region and North Caucasus. We also found a strong correlation



between winter snow accumulation at the WP and observed (GPCC) March precipitation over the North Caucasus, with the highest coefficients (0.5) concentrated close to Elbrus (S9). According to Terskol weather station data, around 30% of the December-March precipitation falls in March, which is likely reflected in the higher share of March snow in cold season accumulation recorded at WP. The snow deposited in March is usually of higher density and is less prone to wind removal.

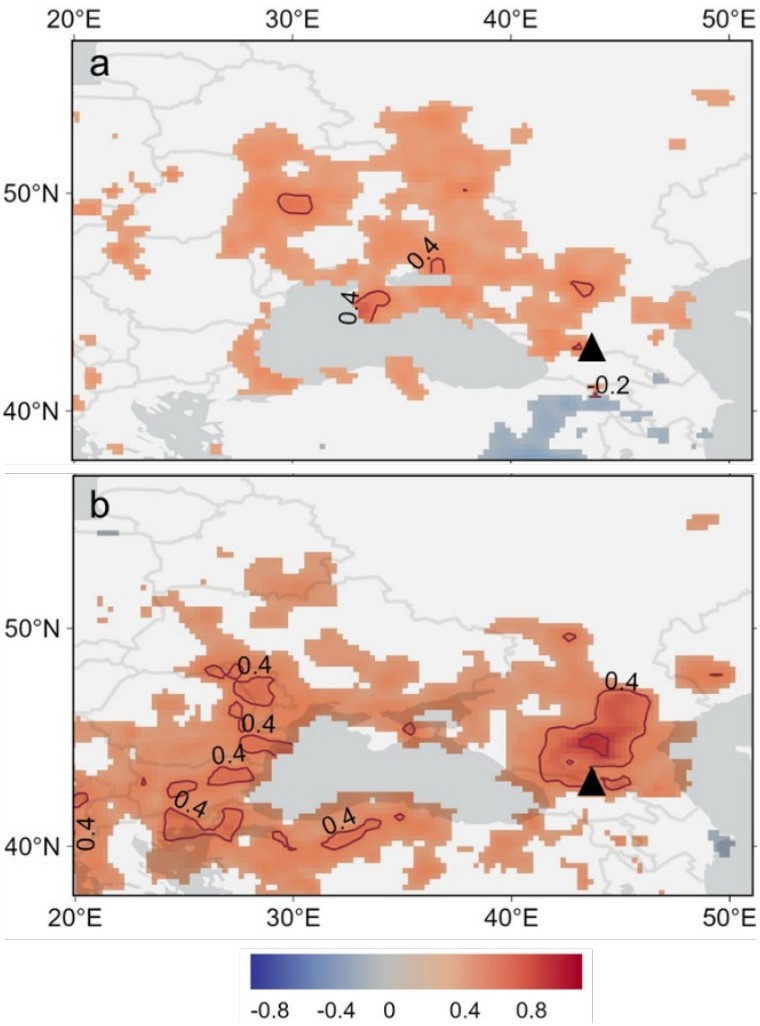

**Figure 8. Spatial correlation of the summer (a) (April-September) and winter (b) (October-March) accumulation with precipitation data of the GPCC v2020 data set in 1950-2009 period. Location of the Elbrus Mt. is shown by black triangle.**

The correlation between seasonal accumulation and gridded precipitation data GPCC v2020 0.25 remains below 0.4 for the majority of the territory. However, during the cold period in a substantial area of the North Caucasus steppe zone, the correlation significantly exceeds 0.4, reaching values ranging from 0.6 to 0.65. A plausible explanation for the strong correlation observed between winter accumulation on the WP and precipitation levels in the foothill areas of the North





Caucasus could be attributed to the relatively frequent occurrence of orographic occlusion. This meteorological phenomenon arises when a wave of cold temperate or Arctic air interacts with the Caucasus Mountains. It commonly transpires in the rear

sector of Mediterranean or Black Sea cyclones, or behind an Atlantic vortex that has shifted from Scandinavia to the Volga region. The presence of the Caucasus ridges impedes the progress of cold air, causing it to circumnavigate the mountains from the direction of the Black and Caspian Seas, thereby saturating the Transcaucasus region. Consequently, the warm and moist air is forced upwards towards the mountains, leading to an extended release of moisture. During summer, this process often manifests in the formation of localized cumulonimbus clouds, while in winter, it results in substantial frontal precipitation

across a significant expanse. These dynamics are most pronounced in the high mountain zone and the foothill areas of the Central Caucasus, offering an explanation for the correlation observed between winter accumulation on the WP and precipitation in the steppe regions of the northern foothills.

### 3.3 Comparison with climate indexes

In the Western and Central Caucasus, the quasi-decadal variability of atmospheric precipitation is linked to the internal

nonlinear dynamics of the climate system, which manifests in the interannual variation of the NAO, AMO, EA/RW, and other indices in temperate latitudes.

The positive precipitation anomaly in 1960-1970 CE period in winter according to the ice-core and meteorological station records in the high mountain Caucasus corresponds to the large-scale pattern of precipitation anomalies. In 1960-1970 CE, a pronounced region of statistically significant precipitation anomalies covered the whole of the Mediterranean and most of the

southern part of Eastern Europe. The value of anomalies reached 15-20 mm/month; in most areas this is 20-30 % of the seasonal amount for the period October-March. These anomalies correspond to pronounced negative geopotential anomalies in the middle troposphere concentrated just over Western Europe. Northern Europe was covered by an extensive precipitation deficit area, which was corresponded to a pronounced positive geopotential anomaly over the North Atlantic. In general, such an anomaly of large-scale circulation agrees well with the negative phase of the NAO and AMO, which prevailed in the cold

seasons of the 1960-1970 CE. The period 2000-2009 CE was characterized by significant inter-annual variability of accumulation according to ice-core data and precipitation at the meteorological stations. However, on averaged over this period, no significant moisture anomalies were detected in Europe.





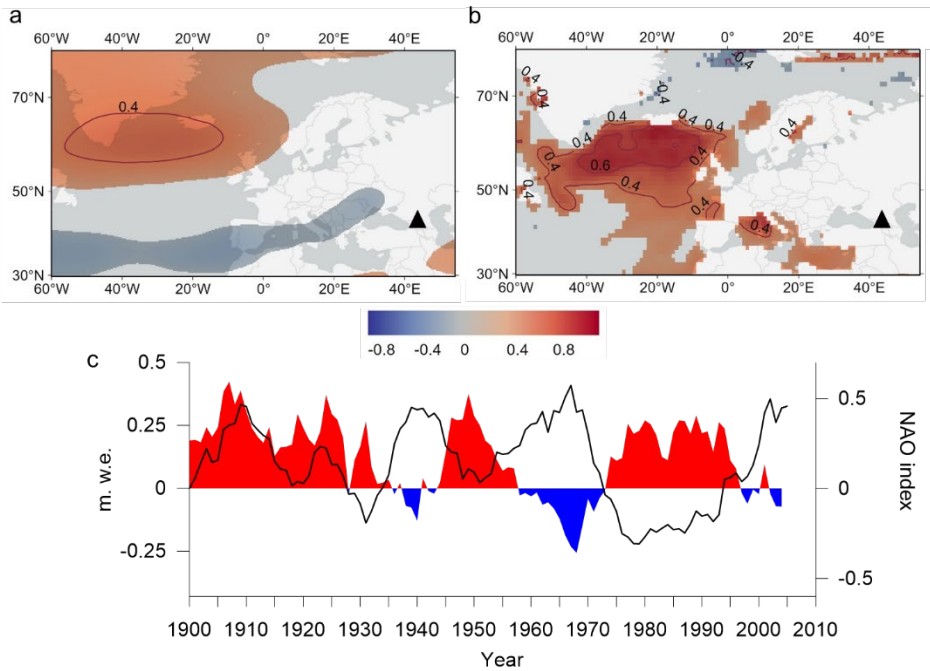

**Figure 9. Spatial correlation of the detrended winter (October-March) annual accumulation at WP with ERA5 500 mb geopotential**
**height data (a) and sea surface temperature (HadlSST1) data in 1950-2009 (b). Ten-year moving average of winter NAO (October-March) and detrended ice core winter accumulation anomalies (c). Location of the Elbrus Mt. is shown by black triangle.**

For the cold period (October-March) in 1930-2009 CE, the 10-year moving averaged WP accumulation anomalies were statistically significantly correlated with the NAO index (r=-0.74, p<0.001) (Fig. 9c). In the Caucasus, negative NAO values
often correspond to an atypical for the North Atlantic weakening of cyclogenesis over Iceland and the formation of anticyclones over the northern Europe. Then, cyclogenesis at the polar front over the Mediterranean and Black Seas becomes more active, which causes positive precipitation anomalies in the Caucasus. The influence of the North Atlantic circulation on the cold season precipitation over the region is highlighted by the correlation of the WP winter accumulation with the Sea surface temperature and geopotential height anomalies in North Atlantic (Fig. 9a,b).

A relationship between precipitation regime and the NAO index has been shown in previous studies (Deser et al., 2017; Vicente-Serrano and López-Moreno, 2008), particularly during winter months. Positive correlation coefficients between NAO and precipitation sums are characteristic for Northern Europe, while negative correlations are observed for the Mediterranean region and southern Europe, which is also evident in our case. Furthermore, a statistically significant inverse relationship between winter precipitation and the NAO index in Turkey (a region relatively close to the Central Caucasus) was revealed in
the study by Türkeş and Ecmel (2005), with a coefficient of -0.4 over a significant portion of the territory. The substantial role of NAO in shaping the precipitation regime of southern Europe and Turkey is demonstrated in the work by (López-Moreno et al., 2011).



Our data suggests that such relationships are not persistent over longer timescales. The moderate positive correlation between NAO and accumulation in cold season was found for the earlier period 1880-1925 (r=0.6, p < 0.001). This period was also characterized by increased summer accumulation and overall precipitation variability. The varying Elbrus accumulation – NAO correlations is in line with previous findings of instabilities in connection of precipitation to large-scale atmospheric circulation at decadal timescales over southern and central Europe (Pauling et al., 2006).

We did not found any significant relationships between summer accumulation records and climate indexes. In summer, the NAO is significantly weaker due to the greater role of mesoscale processes in the formation of seasonal precipitation, as well as the proximity of the Black Sea. In particular, the increase in precipitation and the related rise in summer accumulation during 2000-2009 may be a response to the increase in sea surface temperature in the eastern part of the Black Sea which could be a cause of the observed precipitation increase in southern Russia (Aleshina et al., 2018). The role of positive sea surface temperature anomalies in the Black Sea in the formation of synoptic situations leading to extreme precipitation in the Western Caucasus is clearly demonstrated in the study by (Meredith et al., 2015). Additionally, the increase in summer accumulation in the first decade of the 21st century is generally consistent with the overall positive trend of convective precipitation (Chernokulsky et al., 2019).

Our findings suggest that heat transport in the North Atlantic is of critical importance in determining the precipitation regime in cold season over the Black Sea and Northern Caucasus.

## 3.4 Comparison with paleo records

Reconstructions of the accumulation and precipitation from ice cores from mountain glaciers are relatively limited compared to the polar regions. Thus, in the Alps, the main problem is the influence of avalanche feeding and significant blizzard redistribution of snow at the drilling sites (Bohleber, 2019).

Several estimates of past precipitation were published for the Caucasus region. Martin-Benito et al. (2016) reconstructed May-June precipitation in Transcaucasia since 1752 based on the three ring width. Their reconstruction explains 51.2% of the variability of instrumental data. Verhaegen et al. (2020)) used this reconstruction to calculate the annual amount of precipitation in Terskol. The analysis of precipitation frequency, magnitude, seasonality, as well as the source areas and trajectories of the air masses involved in the Colchis Lowland and near Mt. Elbrus, reveals substantial differences in the precipitation regimes of these regions. Consequently, the reliability of this reconstruction is questionable. Tree-ring width based reconstruction of spring (April–July) precipitation in Crimea over 1620–2002 was published (Solomina et al., 2005). The reconstruction accounts for 37% of the variance in observed precipitation over 1896–1988 although the difference in season duration does not allow the direct comparison with WP accumulation record.

We compared WP accumulation reconstruction with the available monthly resolved paleo-reanalysis EKF400 version 2 2x2 degree resolution, covering the period 1603 to 2003 (Valler et al., 2022). In the EKF400 Version 2, the Kalman filtering technique was utilized to couple an ensemble of atmospheric general circulation models for assimilating early instrumental





observations of temperature, surface pressure, and precipitation, in addition to temperature indices derived from historical documents and tree-ring measurements that are sensitive to temperature and moisture.

We observed a robust correlation between 10-year averaged summer and winter accumulation data and precipitation data from the EKF400 Version 2 (Fig. 10). The region exhibiting a stronger correlation is situated to the south of the Eastern European

plain for both summer and winter seasons. This correlation remains consistently strong throughout the instrumental observational period since 1850 CE. However, prior to 1850 CE, the datasets do not align. The summer paleo precipitation records for the region display an unexpected and unsupported strong negative trend, with a decrease from 170 mm $a^{-1}$ to 120 mm $a^{-1}$. Other records do not corroborate this trend. Similarly, winter records also exhibit discrepancies before 1850, possibly attributed to a decrease in the number of observations included in the EKF400 v.2 dataset. The WP accumulation record

effectively captures the decadal and long-term variability for a larger region during both summer and winter seasons.

The most recent little ice age (LIA) glaciation maximum was observed in the Caucasus in the middle of the 19th century and is well reflected in the dendrochronological data from this region. The general glacier retreat in the North Caucasus began in the late 1840s, which was interrupted 4-5 times in 1860-1880, as well as in the 1910s, 1920s, and 1970-80s (Solomina et al., 2016). According to our results, the period of glacier retreat was preceded by a long period of negative precipitation trend in

summer, which was accompanied by an increase in summer temperature, reconstructed from the tree ring widths (Dolgova, 2016). The period of short readvances at the end of 19[th] century coincide with several intervals of increased summer and winter accumulation. The recent increase in both summer and winter accumulation, despite the continued rise in air temperature, caused the positioning of glacier fronts and their minor advance in the 1970s and 1980s (Solomina et al., 2016).

The primary driver of climatic changes during the late eighteenth and nineteenth centuries was the natural quasi-decadal

variability of the thermohaline ocean circulation. The summer temperatures reconstructed for the Caucasus using tree rings are strongly correlated with AMO oscillations (Dolgova, 2016). Our findings support the role of North Atlantic oscillations not only in temperature, but also in precipitation levels in this region especially during winter seasons.

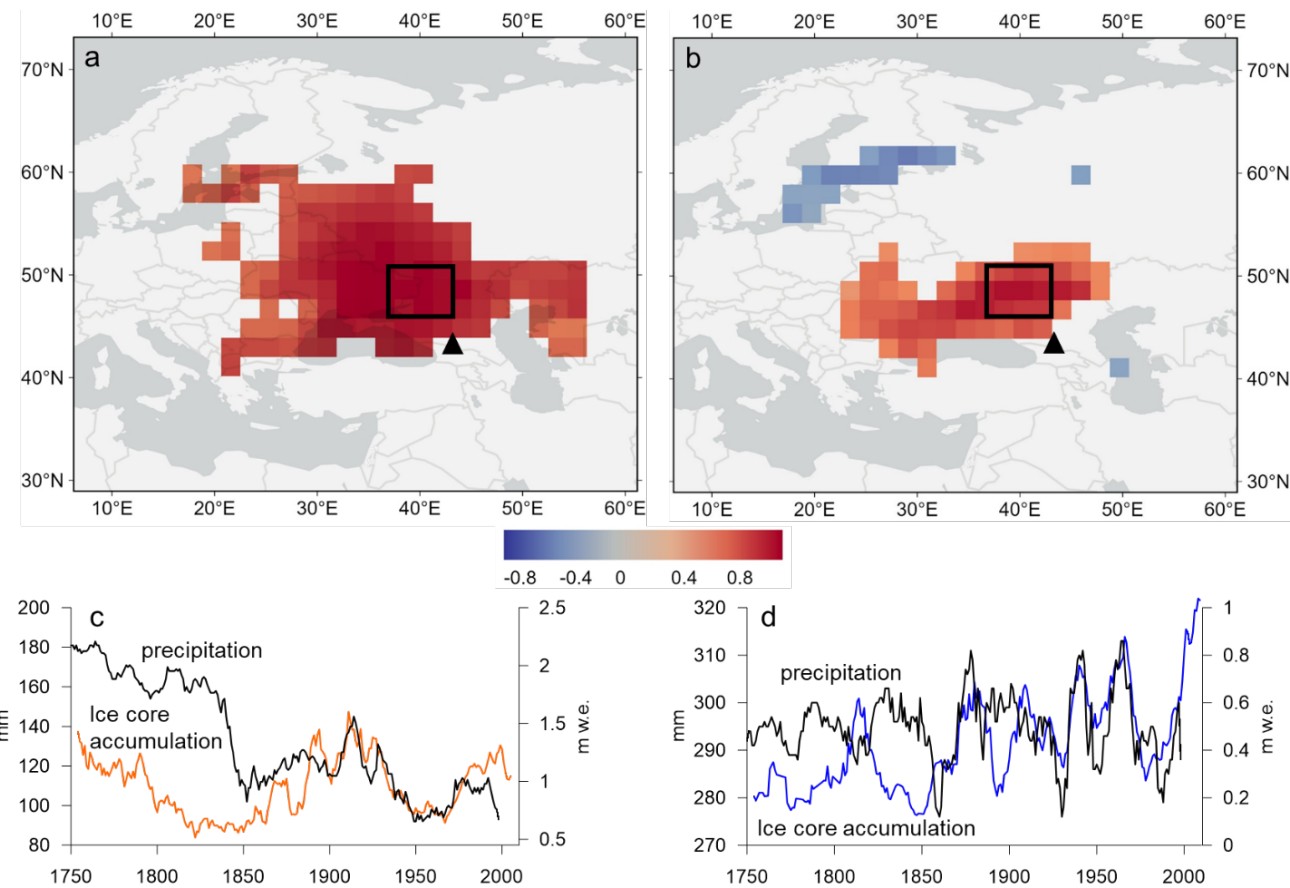

**Figure 10. Spatial correlation of the 10-year moving average of summer (a) (April September) and winter (b) (October-March) WP accumulation with EK400v2 paleo reanalysis ensemble mean precipitation data over 1900-2003 period, time series of the precipitation data averaged over the area indicated by the black box for summer (c) and winter (d) shown together with WP accumulation data. Location of the Elbrus Mt. is shown by black triangle.**

## 4 Conclusions

The ice core retrieved from Mt. Elbrus was dated back to 1750 CE using annual layer counting, with a focus on high-resolution oscillations of ammonium and succinic acid. A new depth-age scale was established by reference horizons associated with known volcanic eruptions. The high accumulation rate and sampling resolution of the ice core facilitated the identification of both annual and seasonal layers. The reconstructed accumulation with the process of layers thinning and the upstream effect accounted for allowed for a meaningful comparison with meteorological data.

To distinguish between winter and summer seasons in meteorological data, a method based on Convective Available Potential Energy was proposed, resulting in a significant improvement in detecting annual and seasonal precipitation. Comparisons of the distribution functions and spectral density of annual precipitation between meteorological stations in the Caucasus and ice-core records provided clear agreement, validating the reconstructed data as representative of precipitation.



The comparison between the obtained ice core accumulation record from WP and gridded precipitation datasets, as well as paleo reanalysis data, revealed that the accumulation record represents the precipitation regime on decadal and long-term

timescales in a larger region encompassing the Northern Caucasus, Black Sea, and South-Eastern Europe.

The analysis showed that precipitation variability in the WP was primarily influenced by the summer season, with a significant decrease in the contribution of summer precipitation over a span of 260 years, corresponding to the transition from the Little Ice Age to modern climatic conditions.

Variations in precipitation were identified with periodicities of 20 and 40 years, corresponding to the typical quasi-decadal

variability associated with North Atlantic thermohaline circulation oscillations. Statistically significant relationship was observed between ice core accumulation pattern and fluctuations in the NAO. Similar variations were found in dendrochronological data on air temperature anomalies in the Caucasus, which aligned with the AMO index. This supports the hypothesis that the quasi-decadal variations in temperature-humidity regime in the Caucasus have an oceanic nature.

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

**Data availability:**

Data are available from the authors upon request. Reanalysis ERA5 data are available at https://cds.climate.copernicus.eu/ (last access 16 May 2023), GPCC data are available at https://psl.noaa.gov/data/gridded/data.gpcc.html (last access: 16 May 2023). Ensemble Kalman Fitting Paleo-Reanalysis Version 2.0 (EKF400_v2.0) is available at https://www.wdc-climate.de/ui/entry?acronym=EKF400_v2.0. (last access 16 May 2023).

**Author contributions:**

The paper was written by VM and SK with contributions from PT, ML, GC, and SS. The ice-core chemistry records were produced by SP and ML; the water isotope measurements were made by AK and VL. The layer-counted timescale was developed by ML, SP, SK, MV and AK. Volcanic identification and synchronization to time scale were performed by ML and SP. Seasonal snow accumulation distribution was estimated by IL. GC calculated backward ice flow trajectories. SS





constructed the thinning function. Interpretation of the accumulation rate history was performed by SK, VM and PT. All the authors read and discussed the manuscript and contributed to improving the final paper.

**Competing interests:**

The authors declare that they have no conflict of interest.

**Acknowledgements:**

This work was carried out within the framework of the Russian Science Foundation project 17-17-01270. The study was completed in the laboratory created within Megagrant project (agreement no. 075-15-2021-599, 08.06.2021). We thank all the people who provided support at all stages of the work, participated in the field work, carried out the sampling and analytical processing of the ice core. We are grateful to V. Matskovsky for his help in interpreting dendrochronological data for the Caucasus.