# Peer review of "Accumulation rates over the past 260 years archived in Elbrus ice core, Caucasus"

_Climate of the Past, 2023_

## Referee Comment (RC1)

This manuscript presents a seasonally resolved accumulation record spanning the period from 1750 to 2009, reconstructed from an ice core from the Elbrus Western Plateau in the Caucasus. The study investigates and discusses dating uncertainty of the ice core archive. It applies ice flow models to correct for layer thinning and to investigate upstream effects to finally derive reconstructed net accumulation rates. Further meteorological station and reanalysis data for the region are investigated and different approaches and methods finally applied to compare those with the reconstruction and previously published/available data from other paleo-archives. The results show, that the ice core based reconstructed accumulation is representative for a large region south of the Eastern European plain and Black sea region with summer precipitation being the primary driver of precipitation variability. A relationship between changes in regional precipitation and fluctuations of the North Atlantic Oscillation index was found, supporting the previous hypothesis that quasi-decadal variations in the temperature-moisture regime of the Caucasus are controlled by oceanic processes. Overall, this is an interesting and enjoyable paper to read and the methods and approaches applied are very original and of high standard. Therefore, I strongly recommend the editor to accept this manuscript for publication in The Cryosphere after some minor revisions outlined in the following.

**General and main comments:**

Unfortunately, there is no explanation how exactly winter and summer has been separated to allow reconstruction of summer and winter accumulation (was there a threshold used in NH4+ concentrations? How was this threshold defined and how were trends in the NH4+ profile data considered since a temporal shift in concentrations would also require a shift of the threshold value over time)? I am aware, that this was already discussed and presented in earlier studies, but because not everyone might be, a very brief summary with a clear and explicit reference to this earlier work should be added. Generally, I found quite a number of inaccuracies in the formulations, which, probably to some extent related to language, should be rather easy to be solved, but caused my review to become much, much longer than anticipated. Again, to be clear, I liked the manuscript a lot!

My four main points are:

1) That despite the statement that winter layers were difficult to determine in the deepest section and reconstruction of the oldest part, thus not very reliable, some of the presented values and in some of the discussion this seems not to fully/always be considered. See statement in Line 274-277: "However, this trend is likely due to the insufficient sampling resolution of the deepest layers, which failed to fully capture the winter layers for a certain year and resulted in an underestimation of winter accumulation below 110 mwe depth (corresponding to the year 1865 CE)." And later on for example in Line 288-290: "The highest percentage of summer accumulation occurred between 1750 and 1830 CE, with an average contribution of 83% to the annual total, resulting in an anomaly of over 200% compared to the modern level in some years." This needs to be more carefully considered and discussed.

2) I somewhat miss an in-depth discussion of the uncertainty of the reconstruction. Ideally also shown graphically (e.g. as a shaded band in the figures). An uncertainty should not be so hard to estimate. Basically the uncertainty range is given by the dating uncertainty plus the correction for the upstream effect for which a reasonable value can easily be derived from the uncertainty of the linear regression applied. This applies to the Result and Discussion Chapter and could be introduced in Section 3.1 (Net accumulation reconstruction). As already mentioned above, for the oldest part, there seems to exist additional uncertainty for the seasonally resolved accumulation reconstructions.

3) I like the approach to calculate and consider the seasons based on CAPE very much (Section 3.2.3 Comparison with gridded data). I found it very convincing and the improvements when applied is evident. Although I can understand that for reasons of simplicity this was not

considered further/latter in the manuscript, I thus still find it a little disappointing. What might be achieved in a reasonable amount of time is to perform similar spatial correlations as performed in the following section for the ice core data. At least for one of the stations closest to the drill site (for example Terskol for which the CAPE corrected results are shown in Fig 7 and the data should thus be readily available). Finding similar patterns as for the ice core, in my opinion, would be an additional, very strong confirmation to show that the ice core reconstructed accumulation is reflecting the regional/local precipitation signal. Please consider this (see details below).

4) The first part of Section 3.3 is completely free of any reference. Either, the authors refer to findings from other studies (in which case they should be cited) or these are their own results in which case I completely miss the context since there is also no reference to any of the applied methods, used data-sets or any figures/tables. I understand that it somehow leads up to the comparison and discussion with the NAO index but I completely miss the context. Please rework this bit.

**Detail comments:**

Line 16: While the application to compare the finally derived seasonal meteorological data with ice core results may be novel, it seems, based on your citations that this approach was used/developed earlier (e.g. Chen et al., 2008; Markowski and Richardson, 2010). I thus suggest replacing "developed" with "applied".

Line 20-23: Do you mean: We identified a statistically significant relationship between the regional changes in precipitation and fluctuations of the North Atlantic Oscillation (NAO) index, which is variable over time.

Line 20-23: "We identified a statistically significant but unstable in time relationship between changes in precipitation in the region and fluctuations of the North Atlantic Oscillation (NAO) index."
This sentence would need some language editing ("…a statistically significant relationship between changes in regional precipitation and fluctuations in the North Atlantic Oscillation (NAO) index which is however unstable over time.") but anyhow, based on your results, a more accurate phrasing seems to be: "We identified a statistically significant relationship between changes in regional precipitation and fluctuations in the North Atlantic Oscillation (NAO) index, which is however not stable over the entire period covered by the ice core." Please consider.

Line 29, 30: "A particularly large mosaic of precipitation records is observed in mountainous areas due to the complex interaction of circulation factors with the underlying surface." This is unclear to me. Do you mean data is sparse or that particularly large, small-scale variations are observed in such regions? Please clarify.

Line 33-35: "Despite its discrepancies they are often used for investigating long-term climatic changes. However, their major limitation is generally coarse spatial resolution which is especially crucial in mountain environments where orographic effects play important role." I assume, they would be used to investigate precipitation changes in particular and not climatic changes in general. Also, especially for the first sentence, a reference would be required. Instead, you might want to consider a reformulation, e.g.: "The discrepancies between these data sets highlights their limitation, and the general difficulty, to investigate long-term precipitation changes. In any case, a major drawback is their generally coarse spatial resolution, which is especially problematic in mountain environments where orographic effects play an important role."

Line 46: "Unlike other proxy, glaciers contain a direct precipitation signal. Annual layer thickness in ice cores depends on total precipitation amount although annual precipitation not always equal to net

accumulation. The most accurate data can be obtained in areas where the snow mass loss due to melting, sublimation, wind and avalanche snow redistribution is minimal." The second sentence contradicts the statement in the first sentence. Maybe better: "Unlike other proxy, glaciers contain a more direct precipitation signal. Annual layer thickness in ice cores depend on the total annual precipitation amount, although the amount of precipitation may not always be equal to net accumulation. Thus, the most accurate data can be obtained in areas where the loss of deposited snow mass due to melt, sublimation and/or erosion and redistribution by wind and avalanches is minimal."

Line 52: "To obtain past accumulation rates, the annual-layer thickness has to be corrected for the cumulative effect of ice flow." To provide some additional information for clarification to non-experts, you may want to change to: "To obtain past accumulation rates, the annual-layer thickness has to be corrected for the cumulative effect of layer thinning with depth, which is caused by ice flow."

Line 52-54: "The algorithm for calculating the initial thickness of deposited annual layers at the surface is quite well developed (Dansgaard and Johnsen, 1969; Nye, 1963; Paterson and Waddington, 1984; Schwerzmann et al., 2006)." I question if finding an algorithm for those calculations really is the important message here. Isn't it that studying and understanding the physical properties of ice and the understanding of ice flow dynamics was the important development? Consequently, this allowed building models to mathematically describe ice flow physics, with these then also being applicable to perform such calculations like deriving the initial ice thickness. Please reformulate accordingly, e.g.: "With the processes of ice flow being well understood, a number of rather simple models and approaches for calculating the initial thickness of deposited annual layers have been developed over the past decades (e.g. Dansgaard and Johnsen, 1969; Nye, 1963; Paterson and Waddington, 1984; Schwerzmann et al., 2006)."

Line 54-56: "The accuracy of accumulation reconstruction depends on the use of ice flow models to estimate the displacement of the drilling site due to the movement of the glacier and the thinning of the annual horizons, especially for the deep parts of the glacier (Licciulli et al., 2019)." This sentence seems not to be a direct citation of what is written in Licciulli et al., 2019. At least I cannot find a similar statement there. I assume that this reference was rather provided because a lot about the basics of ice flow modelling is covered in there. I do not regard this as a problem, but I think estimating the displacement of the drilling site due to the movement of the glacier is not really what is a key message to explain how accumulation is reconstructed. Most relevant seems that the thinning is particularly important for cold glacier sites (ice frozen to bedrock) which creates the shear responsible for the thinning (for a glacier sliding on the bed, thinning will be much less) and that thinning is exponential with depth. In any case, it is not the accuracy of accumulation reconstruction which depends on the use of ice flow models but accumulation can simply not be reconstructed if thinning is not corrected for by the use of an ice flow model. I would suggest reformulating to:
"In order to reconstruct accumulation from the determined thickness of annual layers, an ice flow model is required to correct for the amount of thinning with depth due to the flow of ice (e.g. Winski et al., 2017). This is particularity challenging for the deepest parts where bedrock topography can become an important factor (e.g. Licciulli et al., 2019)"

Winski, D., E. Osterberg, D. Ferris, K. Kreutz, C. Wake, S. Campbell, R. Hawley, S. Roy, S. Birkel, D. Introne and M. Handley, Industrial-age doubling of snow accumulation in the Alaska Range linked to tropical ocean warming, Scientific Reports, 2017, 7(1), 17869. DOI: 10.1038/s41598-017-18022-5.

Line 56-59: "For these reasons detailed ice-core reconstructions of accumulation and precipitation are relatively rare (Dahl-Jensen et al., 1993; Goodwin et al., 2016; Henderson et al., 2006; Pohjola et al., 2002; Winstrup et al., 2019; Yao et al., 2008) compare**d** to other climate and environmental parameters." You should distinguish between reconstructions from Polar and High-elevation ice cores

(for which they are even more sparse) and for the alpine ones, please add a few more references which you might have missed:

Winski, D., E. Osterberg, D. Ferris, K. Kreutz, C. Wake, S. Campbell, R. Hawley, S. Roy, S. Birkel, D. Introne and M. Handley, Industrial-age doubling of snow accumulation in the Alaska Range linked to tropical ocean warming, Scientific Reports, 2017, 7(1), 17869. DOI: 10.1038/s41598-017-18022-5.

Mariani, I., Eichler, A., Jenk, T. M., Brönnimann, S., Auchmann, R., Leuenberger, M. C., and Schwikowski, M.: Temperature and precipitation signal in two Alpine ice cores over the period 1961–2001, Clim. Past, 10, 1093–1108, https://doi.org/10.5194/cp-10-1093-2014, 2014.

Zhang, W., Hou, S., Wu, S.-Y., Pang, H., Sneed, S. B., Korotkikh, E. V., Mayewski, P. A., Jenk, T. M., and Schwikowski, M.: A quantitative method of resolving annual precipitation for the past millennia from Tibetan ice cores, The Cryosphere, 16, 1997–2008, https://doi.org/10.5194/tc-16-1997-2022, 2022.

P.A. Herren, A. Eichler, H. Machguth, T. Papina, L. Tobler, A. Zapf, M. Schwikowski: The onset of Neoglaciation 6000 years ago in western Mongolia revealed by an ice core from the Tsambagarav mountain range Quat. Sci. Rev., 69 (2013), pp. 59-68

Line 73, 74: "A 181.8 m ice core was recovered at the WP in August-September 2009 and the crater in August 2020 (Mikhalenko et al., 2020)." Were both 181.8 m long? Please also indicate here which of the two was used in this study.

Line 80: "The amount of precipitation can be determined as the difference between the measured accumulation layer and the loss caused by sublimation, evaporation, and wind-driven snow redistribution." This might be a language problem, but this seems not correct. Rather: The amount of precipitation can be determined as the sum of the measured thickness of the accumulated layer (in meter water equivalent, corrected for the thinning), sublimation (loss, thus negative in sign), evaporation (negative in sign) and the net amount of snow deposition from wind-driven snow redistribution (which may be negative or positive in sign). What is the difference between sublimation and evaporation in this context? Evaporation from the solid phase is what is defined as sublimation?? What about melt? Do you assume that the meltwater percolates and then refreezes within the same annual layer (thus no net effect)? Please clarify.

Line 88-94: "In winter, the maximum snow accumulation shows a clear shift to the northern and eastern parts of the plateau, where it is limited by the northern ridge and the steep wall of the Western summit of Elbrus. In the southern and western parts of the plateau, absolute minima are observed in the winter accumulation fields, which are likely caused by strong winds during winter (Lavrentiev et al., 2022). The area near the drilling site is characterized by mean values of snow accumulation. The total value of snow loss on the WP is estimated to be about 10% (Mikhalenko, 2020). Although we cannot rule out the higher snow accumulation losses for winter layers at the drilling site." I have difficulties to follow here. If I correctly understand, relocation of winter snow was observed leading to net loss in the more southern and western parts of the plateau and net gain on the northern and eastern parts. Then the important message seems to be that the drill site is roughly in-between these two extremes (loss S and W, gain N and E) and can thus be assumed to be in equilibrium also in winter (no loss and no gain). Correct? But then you further write that the total value of snow loss on the WP is estimated to be about 10%. 10% of the annual amount (consequently more than 10% if considering summer only)? This gets particularly confusing when last it is written that higher snow accumulation losses for winter layers cannot be ruled out. Higher than 10%, higher than net 0 or higher than in summer...??? Please reformulate/rearange this section for clarification.

Line 101: The resolution increased from 10 cm to 5 cm, not decreased.

Line 110: "The very low winter NH4+ levels are related to precipitation of the cold half-year". Please define very low.

Line 121: "By its nature, dating using annual layer counting becomes more uncertain with

depth because identification of winter layers is less straightforward due to the decrease of annual layer thicknesses resulting from glacier ice flow (e.g. Paterson and Waddington, 1984)." By its nature? Isn't it because layers become thinner with depth that sufficient sampling resolution becomes critical to resolve the seasonal variations which is the reason that annual layer counting then becomes more uncertain? As a consequence, a smoothed signal is obtained (at some point even an annual or even multi-annual, decadal etc. average signal). Clearly, this does not only make the detection of winter layers more straight forward but also the summer layers (seasonal resolution becomes impossible). Please reformulate and also comment on the consequences for your summer/winter reconstruction in the lower part of the core.

Line 139: To highlight consider changing to "However, the new dating (1750 CE ± 4 years) is consistent with a Tambora layer..." also language: "....a signal possibly related to the Tambora eruption in a layer located at 118.96 or 119.84 mwe and possibly to Laki in a layer at 124.71 mwe depth". Two signals which could be Tambora? Same size? If one is Tambora, what is the other? Please comment.

Line 149, 150: "The accumulation rate history at Mt. Elbrus can be inferred from depth profiles of annual-layer thicknesses in the WP ice core when corrected for firn densification and thinning of layers due to ice flow." You do not correct for firn densification in your model (would depend e.g. on temperature and accumulation rate). I guess the main point here is that you need to convert the determined annual layer thickness measured in meter into meter water equivalent which is necessary because the used ice flow model (Nye) assumes the incompressibility of ice (which is not the case for firn). Please reformulate accordingly or delete "for firn densification".

Line 156: "Although the annual layer thickness exhibits high variability, the data suggest that layer thinning occurs with increasing depth due to ice flow." This observation is not proof of layer thinning due to ice flow anyhow. It could simply be a very strong change of accumulation over time, which, a priori you might not know while for a cold glacier, based on the physical properties of ice, the thinning with depth is known to occur. I do not think this sentence is needed. Delete.

Line 157: "To determine snow accumulation values, we used a simple J. Nye flow model (Dansgaard and Johnsen, 1969).." Nye flow model instead of J. Nye flow model? Anyway, why do you cite Dansgaard and Johnsen 1969 here if you use the Nye model? The correct reference for the Nye model would be:
Nye, J. F. (1963), Correction factor for accumulation measured by the thickness of the annual layers in an ice sheet, J. Glaciol., 4, 785– 788.

Equation 2: Instead of H – depth of glacier I suggest to use H – glacier thickness. Or ice thickness. Please provide the value used/set for H and the value of the mean accumulation rate derived for your best fit.

Line 164-166: I am not sure if I understand correctly. Please try to reformulate for clarification.

Figure S3: In S3a, the model starts to increase again at depth (from around 1800 back to 1750). This cannot be correct. The layer thickness in the Nye model will always decrease with depth (or age plotted in your case; which I actually suggest to change to depth because in equations 1 and 2, age is not one of the model parameters). Please recheck and correct your calculations. In S3b the increase in accumulation prior to 1800 will become higher as a result. Again wrong reference for the Nye model.

2.3.2 Calculation of backward trajectories: Reading the title I anticipated to read about backward trajectories of air masses. You might want to rename, e.g. to: Correction for the upstream effect

Line 170: "To verify the representativeness of the ice core for the accumulation conditions at the drilling site and account for upstream effect, ..." Do you (can you) really verify that? Maybe just: To account for the upstream effect...?

Line 181: "The model used an approximation of the density profile measured in the borehole..." What do you mean by that? A smoothed profile?

Line 183: "Model simulations show that constant value of the flow rate factor in the firn rheological law has very little effect on the geometry of the backward trajectories and the position of their sources on the glacier surface. Therefore, the rate factor may be chosen arbitrary from a wide range of values." I am not sure if I understand. You mean that using a constant, instead of a non-steady state value has little effect and that the selection of this value within a certain range yields very similar results? Can you be more precise about what "a wide range of values" is (for example 2-8 $MPa^{-3}a^{-1}$)?

Figure S5: There, figure caption for panel c is missing. Please add.

Table 1, header col 3 and 6: "H, m asl" In Equation 2, H is defined as depth of glacier (or glacier/ice thickness as suggested there), which is probably not what is shown here. Use Height (or Altitude?) instead. "Values" better to replace with "Parameters"? Also, "P" is nowhere defined (I guess Precipitation).

Line 235-237: "Ammonium concentration in the ice core is maximum during the period of active convection and the accumulation sum over this period corresponds to precipitation of the warm half of the year." Main point: Dating uncertainty, still split in seasons?? There is no explanation how winter and summer has been separated (threshold? how are trends in the data considered,,temporal shift of threshold required...)

Line 269, 270: "...is 1.641 m.w.e. During this same period, the mean summer and winter accumulations are 1.156 m.w.e and 0.485 m.w.e, respectively." Round to 1 digit. At least the last digit should be omitted but rather 2. The uncertainty of your reconstruction is certainly much bigger than 1 mm and probably rather in the range of a few centimeter (dating uncertainty, uncertainty of summer/winter separation, uncertainty of correction for upstream effect etc!).

Line 271, 272: "Due to dating uncertainties this record is suitable for investigations of decadal, multidecadal and long term regional precipitation variations rather than interpretations of the accumulation for the exact years." Extend the discussion about the uncertainty (see General comments). Please also discuss (extend) the uncertainty for the seasonal resolved records (summer, winter) arising from the criteria to distinguish between winter and summer snow (based on ammonium concentration) and potentially also from the sampling resolution.

Line 274: "A slight positive trend (0.018 m w.e. per decade) was estimated for the annual accumulation over the record, attributable to a general increase in winter accumulation." In the next sentence you contradict this. There you argue that because of sampling resolution fully capturing the winter layers below 110 mwe depth failed, the trend in the winter accumulation is most likely an artefact. The part of the sentence after the comma should be deleted.

Line 278: "The period before 1830 was characterized by increased summer accumulation and annual variability." Increased compared to what? Maybe better: by relatively high accumulation and a high annual variability? Did you do some Time of Emergency statistics to define those years? To me this rather looks like 1820 than 1830.

Line 281: ...around a factor of two...

Line 283: Since around 1980 CE...

Line 284, 285: "The WP site exhibits a seasonal distribution of precipitation typical of the Central Caucasus, with convective precipitation leading to a maximum in the summer months." Please add reference.

Line 285: "The mean share of summer accumulation in the total annual accumulation was 70% (STD=18) over the course of 260 years, which is consistent with current measured precipitation data at weather stations." Since you already compiled all this data from stations etc. in the region, a Figure showing the seasonal distribution of precipitation would be very helpful (can be in the supplement).

Line 287-290: " Over the entire period covered by the core data, there is a statistically significant decrease in the contribution of summer accumulation to the annual total (R2 = 0.6). The highest percentage of summer accumulation occurred between 1750 and 1830 CE, with an average contribution of 83% to the annual total, resulting in an anomaly of over 200% compared to the modern level in some years." Since you previously stated that the reconstructed winter accumulation is questionable in the lower (oldest) part, how sure can you be about this findings being robust or even an artefact? 1750-1830? It looks more like 1750 to around 1800 or 1810 to me. Please define "modern" (what is the reference period) and also "some years" (which years? single years or the average over the period?).

Line 290-291: "In contrast, the lowest percentage of the summer component of the annual accumulation was observed in 1935-1980 (57%)." Looking at the figures, it seems clear that for the period 1980 to the end of the record (2017?) this was even lower?

Line 292: "the average value of which was -25%" -25% compared to what? Which reference period?

Line 315 and line 323: "However, as a correlation coefficient exceeding this value not only has statistical but also physical meaning..." and line 323: "...a physically significant radius.." I do not quite understand how the threshold value of the correlation coefficient at which a "physical meaning" starts to exists was/can be defined? Also what is/defines a physically significant radius? Please elaborate.

Line 326: "Statistical analysis of long-term series of annual precipitation (Sochi, Teberda, Krasnaya Polyana) and reconstructed accumulation on WP reveal high similarity of empirical distribution functions (Fig. 6),..." Krasnaya Polyana is not shown in Figure 6a?

Line 332: "In contrast, ice-core records show a significantly larger range of variability, from a decrease of 80% to an increase of 200%." Not sure how reliable these numbers are. See previous comments regarding uncertainty of reconstruction.

Line 333 ff: "This difference can be attributed to the length of the ice-core data, which spans 260 years, significantly longer than the longest continuous meteorological records. Consequently, the ice-core data captures the full range of precipitation variability in the Caucasus, ranging from extremely dry conditions (one-fifth of annual precipitation) to exceptionally wet conditions (double the annual average)." See comment above. Also, how long is the longest continuous meteorological record used here? Add in bracket.

Figure 6: Panel b and c: The range of the y-axis has to be the same to allow for the visual comparison intended with these panels. Please adjust accordingly (e.g. -150 to 200 %)

Line 349: Change to "...indicates that accumulation at Elbrus is primarily related to precipitation and not significantly affected by post-depositional factors (e.g. melt, wind erosion etc.). At least for temporal averages >5 years, the reconstructed record is dominated by the precipitation signal."

Line 353: "However, our calendar displays considerable inter annual variability." Reference to the according table in the supplement is missing. Add "...(Table S8)".

Line 358: "Therefore, accounting for this factor in determining seasonal and annual precipitation amounts is critical." I do not understand why the annual precipitation amount is affected by this. The annual is the total (sum) over the entire year. Therefore, how the year is split in seasons should not matter as the sum over the full year will result exactly the same? Please comment.

Line 359: "Failure to do so can result in errors exceeding 10% of annual precipitation sums, as illustrated in Figure 7." Annual precipitation sums, not seasonal? Why does this affect the annual sum (see above)? Also, I do not know how I can see (or derive) this 10% error from Figure 7. Would maybe including intercept and slope of the liner regressions help?

Line 350-361: "This figure displays regression relations of the accumulation reconstructed from the core with precipitation amounts at the nearest weather stations for the 1979-2009 CE period." This sentence rather seems to belong to the Figure caption of Figure 7 than in the main text.

Figure 7: Please add the period of the data used (1979-2009?). Also, replace "kern" with core or ice core (in the panels, x-axis labels, itself and accordingly the figure caption.

Line 371, 372: "This is due to the greater spatial homogeneity of the precipitation field in the cold half of the year, which is expressed in larger values of the radii of significant correlation." This is interesting but nowhere shown. Maybe you could add a Figure, similar to Fig 5 but for summer and winter respectively to the supplement?

Line 373-375: "To facilitate further analysis, we have defined the cold period as the period from October to March and the warm period as the period from April to September." I like the approach to calculate and consider the seasons based on CAPE very much. It is very convincing and the improvements in doing looks very clear. Although I can understand that for reasons of simplicity this was not considered further/latter in the manuscript, I still find it a little disappointing. What might be achieved in a reasonable amount of time is to perform similar spatial correlations as performed in the following section for the ice core data (3.2.3 Comparison with gridded data). At least for one of the stations closest to the drill site (for example Terskol for which the CAPE corrected results are shown in Fig 7 and the data should thus be readily available). This would allow to immediately, i.e. visually see how the ice core reconstruction compares with the ("CAPE corrected") meteodata. Finding similar patterns as for the ice core, in my opinion, would be an additional, very strong confirmation to show that the ice core reconstructed accumulation is reflecting the regional/local precipitation signal. As a reference, to help better understand where I am going at, you might want to take a look at Figure 10 in Mariani et al., 2014. Please consider (also see next comment).

Mariani, I., Eichler, A., Jenk, T. M., Brönnimann, S., Auchmann, R., Leuenberger, M. C., and Schwikowski, M.: Temperature and precipitation signal in two Alpine ice cores over the period 1961–2001, Clim. Past, 10, 1093–1108, https://doi.org/10.5194/cp-10-1093-2014, 2014.

3.2.3 Comparison with gridded data (also see comment above): I am lacking a bit of insight in this section. I would assume that precipitation, also in this region is a rather local (or smaller regional) signal, especially close to an orographic barrier like the Caucasus. Therefore, a spatial correlation with a location several thousand km away as found for the summer may not be entirely convincing. The

picture obtained for winter (or the pattern in S9), is what one might expected also for summer, but maybe this is really not the case in this region? I am no expert about the characteristics in this region and therefore, for readers as uneducated as myself, the suggestion made in the comment just before would be extremely helpful for further insight.

Line 404-417: Missing references and I also have a hard time to grasp the context to the rest of the section (see point 4 in my Main Comments).

Line 423: "…the 10-year moving averaged WP accumulation anomalies were statistically significantly correlated with the NAO index…" Not: ….show a statistically significant **negative correlation** with the NAO index…?

Line 424-425: "In the Caucasus, negative NAO values often correspond to an atypical for the North Atlantic weakening of cyclogenesis over Iceland and the formation of anticyclones over the northern Europe." Why in the Caucasus?? Should that not say: "Negative NAO values often correspond to a weakening of cyclogenesis over Iceland and the formation of anticyclones over northern Europe which is atypical for the North Atlantic." ? In the next sentence is where you then describe how this is relevant for the Caucasus region.

Line 435: "The substantial role of NAO in shaping the precipitation regime of southern Europe and Turkey is **was also** demonstrated in the work by (López-Moreno et al., 2011)." Please add a word or two what the cited study relies on (modelling, reanalysis data, paleoarchives,…?).

Line 439-443: This section is hard to read and I am not sure if I understand correctly.
What is your definition of "longer time scales" here? Maybe rephrase and restructure a bit to get the message clear? A suggestion for change could be: "Opposite to the last ~80 years where the correlation is negative, a moderate positive correlation between NAO and accumulation in the cold season was found for the earlier period from 1880-1925 (r=0.6, p < 0.001). This period is also characterized by increased summer accumulation and generally high (annual?) precipitation variability. The variation of the correlation between the reconstructed accumulation for Elbrus and the NAO index is in line with previous findings of instabilities in the connection of precipitation and large-scale atmospheric circulation at decadal timescales observed over southern and central Europe (Pauling et al., 2006)." To avoid repetition, the previous sentence referring to "longer time scales" could then be deleted.

Line 456, 457: Please add references (see comment to Line 56-59). The sentence "Thus, in the Alps, the main problem is the influence of avalanche feeding and significant blizzard redistribution of snow at the drilling sites (Bohleber, 2019)." has to be be deleted. First, the "Thus" makes no sense. Also, you cannot generally equal mountain glaciers with the Alps which is a specific mountain range in Europe. Please note that most ice cores (also in the Alps), at least ice cores from well selected sites, are drilled at locations where avalanches affecting the site can be excluded based on the orography. This is a generalizing but very biased statement. Again, please delete.

464: "Consequently, the reliability of this reconstruction is questionable." This seems to me like a subjective opinion. Please delete.

Line 474, 475: "The region exhibiting a stronger correlation is situated to the south of the Eastern European plain for both summer and winter seasons." This looks fairly consistent with the pattern observed for the analysis visualized in Figure 8 which might be worthwhile to point out here?

Line 474 ff: "However, prior to 1850 CE, the datasets do not align. The summer paleo precipitation

records for the region display an unexpected and unsupported strong negative trend, with a decrease from 170 mm a-1 to 120 mm a-1."
Unexpected and unsupported? Based on what/whom? Please add a reference.
"Other records do not corroborate this trend" Add Reference.
"Similarly, winter records also exhibit discrepancies before 1850, possibly attributed to a decrease in the number of observations included in the EKF400 v.2 dataset. The WP accumulation record effectively captures the decadal and long-term variability for a larger region during both summer and winter seasons."
For winter, you doubt your reconstruction for the older part (hard to find the winter layers due to resolution). Please consider the degree of uncertainty of your own reconstruction and state that some of the discrepancies observed prior to 1850 could probably as likely be due to the uncertainty of your own reconstruction.

Line 518: change to "...temperature-moisture regime..."?

**Language:**

Line 14: seasonally resolved?

Line 15: 181.8 m, delete hyphen between number and m

Line 19, 20: "Reconstructed accumulation is representative for a large region south of **the** Eastern European plain and Black sea region. Summer precipitation was found to be the primary driver of precipitation variability."
Maybe better: "Reconstructed accumulation is representative for a large region south of the Eastern European plain and Black sea region with summer precipitation being the primary driver of precipitation variability."

Line 26, 27: Did you mean: "Unlike for most key climate indicators, which show similar trends in most parts of the world, a strong regional variation in the sign of trend is observed for changes in precipitation amounts (IPCC, 2014)." ?

Line 30: In-situ precipitation measurements...

Line 30: ... particularly for snowfall events.

Line 32, 33: "Currently there are 30 gridded global precipitation data sets  available including gaugebased, satellite-related, and reanalysis data sets (Sun et al., 2018)." ?
Maybe better: "Currently there are 30 gridded global precipitation data sets available, which include data based on gauge measurements, derived from satellites and reanalysis products..."

Line 39: ....but generally suffer from a low temporal resolution...

Line 41: In contrast, reconstructions from tree rings allow for annual resolution and can be calibrated...
-   delete easily; this is relative and also a subjective assessment

Line 49: "...there is a balance between snow wind erosion and accumulation." Consider changing to: "...wind erosion and accumulation of snow is in balance... (or even better: ...is close to equilibrium?)"

Line 67: ..an altitudinal range..

Fig.1 caption: (b) glaciers? I see only one glacier marked...

Line 96: ...at the Institute of Environmental Geosciences in Grenoble (France)...

Line 101: ... down to 168.6 m depth. .... 10 cm for the upper part...

Line 115: ...by an increase in acidity...

Line 117: no **.** before Surprisingly (small)

Line 141: In this study, the new dating will be used as the basis for the accumulation reconstruction.

Line 204: ...dot labels.

Line 207: ... from the 2009 CE drill site.

Line 220: ...used the Global Precipitation Climatology Centre

Line 281: The period from 1935 to 1980 CE was again characterized by relatively low accumulation in summer and relatively high accumulation in winter.

Line 429: ....height anomalies over the North Atlantic

Line 434: ".... ( relatively close to the Central Caucasus) " I guess this can be deleted

Line 468: ... EKF400 version 2 with a 2x2 degree...

Line 502: ...the process of layer thinning...  delete the s in layers

Line 505: ....Energy was  applied, resulting...   Why proposed? It was applied.

Line 517: This study supports....

---

## Author Comment (AC1)

**Reply to reviewers' comments on "History of desert dust deposition recorded in the Elbrus ice core"**

*We would like to thank both reviewers for their comments that help us to improve and clarify the manuscript.*

This manuscript presents a seasonally resolved accumulation record spanning the period from 1750 to 2009, reconstructed from an ice core from the Elbrus Western Plateau in the Caucasus. The study investigates and discusses dating uncertainty of the ice core archive. It applies ice flow models to correct for layer thinning and to investigate upstream effects to finally derive reconstructed net accumulation rates. Further meteorological station and reanalysis data for the region are investigated and different approaches and methods finally applied to compare those with the reconstruction and previously published/available data from other paleo-archives. The results show, that the ice core based reconstructed accumulation is representative for a large region south of the Eastern European plain and Black sea region with summer precipitation being the primary driver of precipitation variability. A relationship between changes in regional precipitation and fluctuations of the North Atlantic Oscillation index was found, supporting the previous hypothesis that quasi-decadal variations in the temperature-moisture regime of the Caucasus are controlled by oceanic processes. Overall, this is an interesting and enjoyable paper to read and the methods and approaches applied are very original and of high standard. Therefore, I strongly recommend the editor to accept this manuscript for publication in The Cryosphere after some minor revisions outlined in the following.

*We thank reviewer for these general comments and especially for this extensive and thoughtful review, which helped us to improve the clarity and contents of the manuscript!*

**General and main comments:**
Unfortunately, there is no explanation how exactly winter and summer has been separated to allow reconstruction of summer and winter accumulation (was there a threshold used in NH4+ concentrations? How was this threshold defined and how were trends in the NH4+ profile data considered since a temporal shift in concentrations would also require a shift of the threshold value over time)? I am aware, that this was already discussed and presented in earlier studies, but because not everyone might be, a very brief summary with a clear and explicit reference to this earlier work should be added. Generally, I found quite a number of inaccuracies in the formulations, which, probably to some extent related to language, should be rather easy to be solved, but caused my review to become much, much longer than anticipated. Again, to be clear, I liked the manuscript a lot!

*Taken into account. We added additional information about annual dating criteria and revised this section. "Specific thresholds were used to differentiate between winter and summer conditions in the upper ice layers, reaching down to a depth of 75.6 meters, corresponding to the year 1963 (Mikhalenko et al., 2015). These thresholds were set at 100 ppb for ammonium and 5 ppb for succinate concentrations. To account for the observed increase in ammonium concentrations during the industrial era, these criteria were adjusted. Specifically, between depths of 75.6 to 86.8 meters, covering the period from 1950 to 1963, the ammonium winter criteria were modified to 50 ppb, and further reduced to 30 ppb below that depth. In contrast, no substantial depth-related variation was observed for succinate, and the concentration limit of 5 ppb remained consistent in deeper layers. Below a depth of approximately 150 meters, winter layers often consist of only one or two samples, while summer layers consist of more than six samples, making the process of annual dating challenging (Preunkert et al., 2019)."*

My four main points are:
1) That despite the statement that winter layers were difficult to determine in the deepest section and reconstruction of the oldest part, thus not very reliable, some of the presented values and in some of the discussion this seems not to fully/always be considered. See statement in Line 274-277: "However, this trend is likely due to the insufficient sampling resolution of the deepest layers, which failed to fully capture the winter layers for a certain year and resulted in an underestimation of winter accumulation below 110 mwe depth (corresponding to the year 1865 CE)." And later on for example in Line 288-290: "The highest

percentage of summer accumulation occurred between 1750 and 1830 CE, with an average contribution of 83% to the annual total, resulting in an anomaly of over 200% compared to the modern level in some years." This needs to be more carefully considered and discussed.

*Taken into account. Text revised. Indeed, we cannot be certain about summer/winter precipitation due to insufficient sampling resolution. We revised the text accordingly and only consider 1865-2009 for analysis of the seasonal shares of accumulation.*

2) I somewhat miss an in-depth discussion of the uncertainty of the reconstruction. Ideally also shown graphically (e.g. as a shaded band in the figures). An uncertainty should not be so hard to estimate. Basically the uncertainty range is given by the dating uncertainty plus the correction for the upstream effect for which a reasonable value can easily be derived from the uncertainty of the linear regression applied. This applies to the Result and Discussion Chapter and could be introduced in Section 3.1 (Net accumulation reconstruction). As already mentioned above, for the oldest part, there seems to exist additional uncertainty for the seasonally resolved accumulation reconstructions.

*Taken into account. Text revised. We added a paragraph on the uncertainty and modified figures accordingly.*

*"The uncertainty range for the reconstructed accumulation values is determined by dating uncertainty and correction for the upstream effect. Additionally, for winter accumulation, the uncertainty increases due to lower sampling resolution. The dating accuracy of the reconstruction varies, with a precision of ±1 year for the years 2009-1912 CE, ±2 years in the period 1912-1825 CE, and a decrease to ±4 years in the period 1825-1750 CE. The uncertainty associated with the upstream correction is challenging to estimate precisely, as it depends on the initial uncertainty of the spatial accumulation distribution and assumptions regarding its persistence over time, as well as back trajectory modeling uncertainty. To account for the possible associated errors we estimated the uncertainty of the applied linear regression. Additionally we added uncertainty of 30% of winter accumulation for the period prior 1865 CE where the sampling resolution is not sufficient and is likely largely underestimated."*

3) I like the approach to calculate and consider the seasons based on CAPE very much (Section 3.2.3 Comparison with gridded data). I found it very convincing and the improvements when applied is evident. Although I can understand that for reasons of simplicity this was not considered further/latter in the manuscript, I thus still find it a little disappointing. What might be achieved in a reasonable amount of time is to perform similar spatial correlations as performed in the following section for the ice core data. At least for one of the stations closest to the drill site (for example Terskol for which the CAPE corrected results are shown in Fig 7 and the data should thus be readily available). Finding similar patterns as for the ice core, in my opinion, would be an additional, very strong confirmation to show that the ice core reconstructed accumulation is reflecting the regional/local precipitation signal. Please consider this (see details below).

*Taken into account. We added text and supplementary information as suggested.*
*Terskol precipitation data show similar spatial correlation patterns for seasons and individual months. It's likely that the monthly shares of total precipitation may differ at Terskol and drilling site.*

4) The first part of Section 3.3 is completely free of any reference. Either, the authors refer to findings from other studies (in which case they should be cited) or these are their own results in which case I completely miss the context since there is also no reference to any of the applied methods, used data-sets or any figures/tables. I understand that it somehow leads up to the comparison and discussion with the NAO index but I completely miss the context. Please rework this bit.

*Taken into account. Text revised. We also added one figure to Supplementary.*

**Detail comments:**

Line 16: While the application to compare the finally derived seasonal meteorological data with ice core results may be novel, it seems, based on your citations that this approach was used/developed earlier (e.g. Chen et al., 2008; Markowski and Richardson, 2010). I thus suggest replacing "developed" with "applied".

*Done*

Line 20-23: Do you mean: We identified a statistically significant relationship between the regional changes in precipitation and fluctuations of the North Atlantic Oscillation (NAO) index, which is variable over time.

*Yes, text revised.*

Line 20-23: "We identified a statistically significant but unstable in time relationship between changes in precipitation in the region and fluctuations of the North Atlantic Oscillation (NAO) index."
This sentence would need some language editing ("…a statistically significant relationship between changes in regional precipitation and fluctuations in the North Atlantic Oscillation (NAO) index which is however unstable over time.") but anyhow, based on your results, a more accurate phrasing seems to be: "We identified a statistically significant relationship between changes in regional precipitation and fluctuations in the North Atlantic Oscillation (NAO) index, which is however not stable over the entire period covered by the ice core." Please consider.

*Done, text revised.*

Line 29, 30: "A particularly large mosaic of precipitation records is observed in mountainous areas due to the complex interaction of circulation factors with the underlying surface." This is unclear to me. Do you mean data is sparse or that particularly large, small-scale variations are observed in such regions? Please clarify.

*Done, text revised*

Line 33-35: "Despite its discrepancies they are often used for investigating long-term climatic changes. However, their major limitation is generally coarse spatial resolution which is especially crucial in mountain environments where orographic effects play important role." I assume, they would be used to investigate precipitation changes in particular and not climatic changes in general. Also, especially for the first sentence, a reference would be required. Instead, you might want to consider a reformulation, e.g.: "The discrepancies between these data sets highlights their limitation, and the general difficulty, to investigate long-term precipitation changes. In any case, a major drawback is their generally coarse spatial resolution, which is especially problematic in mountain environments where orographic effects play an important role."

*Done, text revised*

Line 46: "Unlike other proxy, glaciers contain a direct precipitation signal. Annual layer thickness in ice cores depends on total precipitation amount although annual precipitation not always equal to net accumulation. The most accurate data can be obtained in areas where the snow mass loss due to melting, sublimation, wind and avalanche snow redistribution is minimal." The second sentence contradicts the statement in the first sentence. Maybe better: "Unlike other proxy, glaciers contain a more direct precipitation signal. Annual layer thickness in ice cores depend on the total annual precipitation amount, although the amount of precipitation may not always be equal to net accumulation. Thus, the most accurate data can be obtained in areas where the loss of deposited snow mass due to melt, sublimation and/or erosion and redistribution by wind and avalanches is minimal."

*Done, text revised*

Line 52: "To obtain past accumulation rates, the annual-layer thickness has to be corrected for the cumulative effect of ice flow." To provide some additional information for clarification to non-experts, you may want to change to: "To obtain past accumulation rates, the annual-layer thickness has to be corrected for the cumulative effect of layer thinning with depth, which is caused by ice flow."

*Done, text revised*

Line 52-54: "The algorithm for calculating the initial thickness of deposited annual layers at the surface is quite well developed (Dansgaard and Johnsen, 1969; Nye, 1963; Paterson and Waddington, 1984; Schwerzmann et al., 2006)." I question if finding an algorithm for those calculations really is the important message here. Isn't it that studying and understanding the physical properties of ice and the understanding of ice flow dynamics was the important development? Consequently, this allowed building models to mathematically describe ice flow physics, with these then also being applicable to perform such calculations like deriving the initial ice thickness. Please reformulate accordingly, e.g.: "With the processes of ice flow being well understood, a number of rather simple models and approaches for calculating the initial thickness of deposited annual layers have been developed over the past decades (e.g. Dansgaard and Johnsen, 1969; Nye, 1963; Paterson and Waddington, 1984; Schwerzmann et al., 2006)."

*Done, text revised*

Line 54-56: "The accuracy of accumulation reconstruction depends on the use of ice flow models to estimate the displacement of the drilling site due to the movement of the glacier and the thinning of the annual horizons, especially for the deep parts of the glacier (Licciulli et al., 2019)." This sentence seems not to be a direct citation of what is written in Licciulli et al., 2019. At least I cannot find a similar statement there. I assume that this reference was rather provided because a lot about the basics of ice flow modelling is covered in there. I do not regard this as a problem, but I think estimating the displacement of the drilling site due to the movement of the glacier is not really what is a key message to explain how accumulation is reconstructed. Most relevant seems that the thinning is particularly important for cold glacier sites (ice frozen to bedrock) which creates the shear responsible for the thinning (for a glacier sliding on the bed, thinning will be much less) and that thinning is exponential with depth. In any case, it is not the accuracy of accumulation reconstruction which depends on the use of ice flow models but accumulation can simply not be reconstructed if thinning is not corrected for by the use of an ice flow model. I would suggest reformulating to:
"In order to reconstruct accumulation from the determined thickness of annual layers, an ice flow model is required to correct for the amount of thinning with depth due to the flow of ice (e.g. Winski et al., 2017). This is particularity challenging for the deepest parts where bedrock topography can become an important factor (e.g. Licciulli et al., 2019)"
Winski, D., E. Osterberg, D. Ferris, K. Kreutz, C. Wake, S. Campbell, R. Hawley, S. Roy, S. Birkel, D. Introne and M.

Handley, Industrial-age doubling of snow accumulation in the Alaska Range linked to tropical ocean warming, Scientific

Reports, 2017, 7(1), 17869. DOI: 10.1038/s41598-017-18022-5.

*Done, text revised*

Line 56-59: "For these reasons detailed ice-core reconstructions of accumulation and precipitation are relatively rare (Dahl-Jensen et al., 1993; Goodwin et al., 2016; Henderson et al., 2006; Pohjola et al., 2002; Winstrup et al., 2019; Yao et al., 2008) compare**d** to other climate and environmental parameters." You should distinguish between reconstructions from Polar and High-elevation ice cores (for which they are even more sparse) and for the alpine ones, please add a few more references which you might have missed:
Winski, D., E. Osterberg, D. Ferris, K. Kreutz, C. Wake, S. Campbell, R. Hawley, S. Roy, S. Birkel, D. Introne and M.

Handley, Industrial-age doubling of snow accumulation in the Alaska Range linked to tropical ocean warming, Scientific

Reports, 2017, 7(1), 17869. DOI: 10.1038/s41598-017-18022-5.

Mariani, I., Eichler, A., Jenk, T. M., Brönnimann, S., Auchmann, R., Leuenberger, M. C., and Schwikowski, M.:

Temperature and precipitation signal in two Alpine ice cores over the period 1961–2001, Clim. Past, 10, 1093–1108,

https://doi.org/10.5194/cp-10-1093-2014, 2014.

Zhang, W., Hou, S., Wu, S.-Y., Pang, H., Sneed, S. B., Korotkikh, E. V., Mayewski, P. A., Jenk, T. M., and Schwikowski,

M.: A quantitative method of resolving annual precipitation for the past millennia from Tibetan ice cores, The Cryosphere,

16, 1997–2008, https://doi.org/10.5194/tc-16-1997-2022, 2022.

P.A. Herren, A. Eichler, H. Machguth, T. Papina, L. Tobler, A. Zapf, M. Schwikowski: The onset of Neoglaciation 6000

years ago in western Mongolia revealed by an ice core from the Tsambagarav mountain range Quat. Sci. Rev., 69 (2013), pp.

59-68

*Done, text revised, references added*

Line 73, 74: "A 181.8 m ice core was recovered at the WP in August-September 2009 and the crater in August 2020 (Mikhalenko et al., 2020)." Were both 181.8 m long? Please also indicate here which of the two was used in this study.

*Done, text revised*

Line 80: "The amount of precipitation can be determined as the difference between the measured accumulation layer and the loss caused by sublimation, evaporation, and wind-driven snow redistribution." This might be a language problem, but this seems not correct. Rather: The amount of precipitation can be determined as the sum of the measured thickness of the accumulated layer (in meter water equivalent, corrected for the thinning), sublimation (loss, thus negative in sign), evaporation (negative in sign) and the net amount of snow deposition from wind-driven snow redistribution (which may be negative or positive in sign). What is the difference between sublimation and evaporation in this context? Evaporation from the solid phase is what is defined as sublimation?? What about melt? Do you assume that the meltwater percolates and then refreezes within the same annual layer (thus no net effect)? Please clarify.

*Done, text revised. Indeed, it was an inaccurate translation of the physical terms. Terms were clarified and sublimation differentiated from the opposite process of deposition of ice crystals from the water vapor.*

*"The amount of precipitation can be determined as the difference between the measured accumulation layer and the loss caused by melting, sublimation, and wind-driven snow redistribution. Ice formation at the drilling site occurs predominantly in cold, dry conditions and the thickness of the infiltration ice layers, which do not form every year, does not exceed 10 mm (Mikhalenko et al., 2015). The contribution of sublimation rate to glacier mass balance and snow cover balance is estimated to be between 5-10% (Bintanja, 1998; Palm et al., 2017) but can reach up to 30% in certain climatic conditions (Pomeroy and Gray, 1995). On the WP in 2018, sublimation during the short season of possible surface melting was estimated to be 3% of the accumulation layer or 45 mm w.e a-1 (Mikhalenko et al., 2015)."*

Line 88-94: "In winter, the maximum snow accumulation shows a clear shift to the northern and eastern parts of the plateau, where it is limited by the northern ridge and the steep wall of the Western summit of Elbrus. In the southern and western parts of the plateau, absolute minima are observed in the winter accumulation fields, which are likely caused by strong winds during winter (Lavrentiev et al., 2022). The area near the drilling site is characterized by mean values of snow accumulation. The total value of snow loss on the WP is estimated to be about 10% (Mikhalenko, 2020). Although we cannot rule out the higher snow accumulation losses for winter layers at the drilling site." I have difficulties to follow here. If I correctly understand, relocation of winter snow was observed leading to net loss in the more southern and western parts of the plateau and net gain on the northern and eastern parts. Then the important message seems to be that the drill site is roughly in-between these two extremes (loss S and W, gain N and E) and can thus be assumed to be in equilibrium also in winter (no loss and no gain). Correct? But then you further write that the total value of snow loss on the WP is estimated to be about 10%. 10% of the annual amount (consequently more than 10% if considering summer only)? This gets particularly confusing when last it is written that higher snow accumulation losses for winter layers cannot be ruled out. Higher than 10%, higher than net 0 or higher than in summer…??? Please reformulate/rearange this section for clarification.

*Done, text revised.*
*Wind-driven snow redistribution was measured by stakes on the plateau during three field seasons, showing a zero balance between scouring erosion and accumulation of snow near the drilling site (Mikhalenko, 2020). An analysis of the fields of summer and winter accumulation on the plateau from 2015-2017 shows that snow accumulation in the summer period is evenly distributed over the plateau. The relocation of winter snow was observed leading to net loss in the southern and western parts of the plateau and net gain on the northern and eastern parts of the plateau limited by the northern ridge and the steep wall of the Western summit of Elbrus (Lavrentiev et al., 2022). The drill site is roughly in-between these two extremes and can thus be assumed to be in equilibrium also in winter. While we cannot dismiss the possibility of snow accumulation losses driven by wind for winter layers in specific years at the drilling site."*

Line 101: The resolution increased from 10 cm to 5 cm, not decreased.
*Done, text revised.*

Line 110: "The very low winter NH4+ levels are related to precipitation of the cold half-year". Please define very low.

As suggested earlier we revised text and added more details on dating methodology.
*"Specific thresholds were used to differentiate between winter and summer conditions in the upper ice layers, reaching down to a depth of 75.6 meters, corresponding to the year 1963 (Mikhalenko et al., 2015). These thresholds were set at 100 ppb for ammonium and 5 ppb for succinate concentrations. To account for the observed increase in ammonium concentrations during the industrial era, these criteria were adjusted. Specifically, between depths of 75.6 to 86.8 meters, covering the period from 1950 to 1963, the ammonium winter criteria were modified to 50 ppb, and further reduced to 30 ppb below that depth. In contrast, no substantial depth-related variation was observed for succinate, and the concentration limit of 5 ppb remained consistent in deeper layers. Below a depth of approximately 150 meters, winter layers often consist of only one or two samples, while summer layers consist of more than six samples, making the process of annual dating challenging (Preunkert et al., 2019)."*

Line 121: "By its nature, dating using annual layer counting becomes more uncertain with depth because identification of winter layers is less straightforward due to the decrease of annual layer thicknesses resulting from glacier ice flow (e.g. Paterson and Waddington, 1984)." By its nature? Isn't it because layers become thinner with depth that sufficient sampling resolution becomes critical to resolve the seasonal variations which is the reason that annual layer counting then becomes more uncertain? As a consequence, a smoothed signal is obtained (at some point even an annual or even multi-annual, decadal etc. average signal). Clearly, this does not only make the detection of winter layers more straight forward but also the summer layers (seasonal resolution becomes impossible). Please reformulate and also comment on the consequences for your summer/winter reconstruction the lower part of the core.

*Done. We included more information on sampling resolution and challenging winter layer identification from a certain depth/age. In the discussion, we revised the approach and highlighted that quantitative winter accumulation reconstruction is only reliable since 1865. We also added this information to figures.*

Line 139: To highlight consider changing to "However, the new dating (1750 CE ± 4 years) is consistent with a Tambora layer…" also language: "….a signal possibly related to the Tambora eruption in a layer located at 118.96 or 119.84 mwe and possibly to Laki in a layer at 124.71 mwe depth". Two signals which could be Tambora? Same size? If one is Tambora, what is the other? Please comment.

*Previous studies showed a double peak in Greenland and Antarctica (in 1816 supposed to be Tambora and another one in 1810 (still unknown origin)). So we cannot decide without tephra analysis.*

*Henrik B. Clausen, Claus U. Hammer, Christine S. Hvidberg, Dorthe Dahl-Jensen, Jørgen P. Steffensen, Josef Kipfstuhl, Michel Legrand. JGR. A comparison of the volcanic records over the past 4000 years from the Greenland Ice Core Project and Dye 3 Greenland ice cores https://doi.org/10.1029/97JC00587.*

Line 149, 150: "The accumulation rate history at Mt. Elbrus can be inferred from depth profiles of annual-layer thicknesses in the WP ice core when corrected for firn densification and thinning of layers due to ice flow." You do not correct for firn densification in your model (would depend e.g. on temperature and accumulation rate). I guess the main point here is that you need to convert the determined annual layer thickness measured in meter into meter water equivalent which is necessary because the used ice flow model (Nye) assumes the incompressibility of ice (which is not the case for firn). Please reformulate accordingly or delete "for firn densification".

*Done. Text revised*

Line 156: "Although the annual layer thickness exhibits high variability, the data suggest that layer thinning occurs with increasing depth due to ice flow." This observation is not proof of layer thinning due to ice flow anyhow. It could simply be a very strong change of accumulation over time, which, a priori you might not know while for a cold glacier, based on the physical properties of ice, the thinning with depth is known to occur. I do not think this sentence is needed. Delete.

*Done. Removed*

Line 157: "To determine snow accumulation values, we used a simple J. Nye flow model (Dansgaard and Johnsen, 1969).." Nye flow model instead of J. Nye flow model? Anyway, why do you cite Dansgaard and Johnsen 1969 here if you use the Nye model? The correct reference for the Nye model would be:
Nye, J. F. (1963), Correction factor for accumulation measured by the
thickness of the annual layers in an ice sheet, J. Glaciol., 4, 785– 788.
Equation 2: Instead of H – depth of glacier I suggest to use H – glacier thickness. Or ice thickness. Please provide the value used/set for H and the value of the mean accumulation rate derived for your best fit.

*Done. Reference added, values added. Although we used introduced by Nye but time-integrated and further explained by Dansgaard and Johnsen 1969.*

Line 164-166: I am not sure if I understand correctly. Please try to reformulate for clarification.

*Done, part of the text removed.*

Figure S3: In S3a, the model starts to increase again at depth (from around 1800 back to 1750). This cannot be correct. The layer thickness in the Nye model will always decrease with depth (or age plotted in your case; which I actually suggest to change to depth because in equations 1 and 2, age is not one of the model parameters). Please recheck and correct your calculations. In S3b the increase in accumulation prior to 1800 will become higher as a result. Again wrong reference for the Nye model.

*Done. Thank you for noticing it. We now placed correct figures although similarly to Dansgaard and Johnsen 1969 we show layers thickness vs age in yrs.*

2.3.2 Calculation of backward trajectories: Reading the title I anticipated to read about backward trajectories of air masses. You might want to rename, e.g. to: Correction for the upstream effect

*Done. Text revised.*

Line 170: "To verify the representativeness of the ice core for the accumulation conditions at the drilling site and account for upstream effect, …" Do you (can you) really verify that? Maybe just: To account for the upstream effect...?

*Done. Text revised.*

Line 181: "The model used an approximation of the density profile measured in the borehole…" What do you mean by that? A smoothed profile?

*We mean an approximation by smoothing of the depth-density distribution inferred from the ice core. Text revised.*

Line 183: "Model simulations show that constant value of the flow rate factor in the firn rheological law has very little effect on the geometry of the backward trajectories and the position of their sources on the glacier surface. Therefore, the rate factor may be chosen arbitrary from a wide range of values." I am not sure if I understand. You mean that using a constant, instead of a non-steady state value has little effect and that the selection of this value within a certain range yields very similar results? Can you be more precise about what "a wide range of values" is (for example 2-8 MPa$_{-3a-1}$)?

*Here, we do not compare how constant and non-steady state rate factors influence the result of the simulation, as we consider only steady-state case. Indeed, the selection of this value within a certain range yields very similar results. We have rewritten this part of the article to avoid the ambiguity and specified the range of the rate factor values (from 4.82 MPa–3a–1 to 14.46 MPa–3a–1).*

Figure S5: There, figure caption for panel c is missing. Please add.

*Done.*

Table 1, header col 3 and 6: "H, m asl" In Equation 2, H is defined as depth of glacier (or glacier/ice thickness as suggested there), which is probably not what is shown here. Use Height (or Altitude?) instead. "Values" better to replace with "Parameters"? Also, "P" is nowhere defined (I guess Precipitation).

*Done. Table revised.*

Line 235-237: "Ammonium concentration in the ice core is maximum during the period of active convection and the accumulation sum over this period corresponds to precipitation of the warm half of the year." Main point: Dating uncertainty, still split in seasons?? There is no explanation how winter and summer has been separated (threshold? how are trends in the data considered,,temporal shift of threshold required...)

*We added detailed discussion on dating and season separation earlier in the text.*

Line 269, 270: "…is 1.641 m.w.e. During this same period, the mean summer and winter accumulations are 1.156 m.w.e and 0.485 m.w.e, respectively." Round to 1 digit. At least the last digit should be omitted but rather 2. The uncertainty of your reconstruction is certainly much bigger than 1 mm and probably rather in the range of a few centimeter (dating uncertainty, uncertainty of summer/winter separation, uncertainty of correction for upstream effect etc!).

*Done. We agree with this comment. Text revised accordingly.*

Line 271, 272: "Due to dating uncertainties this record is suitable for investigations of decadal, multidecadal and long term regional precipitation variations rather than interpretations of the accumulation for the exact years." Extend the discussion about the uncertainty (see General comments). Please also discuss (extend) the uncertainty for the seasonal resolved records (summer, winter) arising from the criteria to distinguish between winter and summer snow (based on ammonium concentration) and potentially also from the sampling resolution.

*Taken into account. Text revised. We now begin this section with the uncertainty discussion. We also changes figure 4 and included shaded filled areas to illustrate uncertainty.*

*"The graphical results of the accumulation reconstruction are presented in Figure 4. The uncertainty range for the reconstructed accumulation values is determined by dating uncertainty and correction for the upstream effect. Additionally, for winter accumulation, the uncertainty increases due to lower sampling resolution. The dating accuracy of the reconstruction varies, with a precision of ±1 year for the years 2009-1912 CE, ±2 years in the period 1912-1825 CE, and a decrease to ±4 years in the period 1825-1750 CE The uncertainty associated with the upstream correction is challenging to estimate precisely, as it depends on the initial uncertainty of the spatial accumulation distribution and assumptions regarding its persistence over time, as well as uncertainty in back trajectory modeling. To account for the potential associated errors, we estimated the uncertainty of the applied linear regression. Additionally, we added an uncertainty of 30% of winter accumulation for the period prior to 1865 CE, where the sampling resolution is insufficient and likely underestimated."*

Line 274: "A slight positive trend (0.018 m w.e. per decade) was estimated for the annual accumulation over the record, attributable to a general increase in winter accumulation." In the next sentence you contradict this. There you argue that because of sampling resolution fully capturing the winter layers below 110 mwe depth failed, the trend in the winter accumulation is most likely an artefact. The part of the sentence after the comma should be deleted.

*Taken into account. Text revised. Now we just stated that there is no statistically significant trend in accumulation reconstructions.*

Line 278: "The period before 1830 was characterized by increased summer accumulation and annual

variability." Increased compared to what? Maybe better: by relatively high accumulation and a high annual variability? Did you do some Time of Emergency statistics to define those years? To me this rather looks like 1820 than 1830.

*Taken into account. Text revised.*

Line 281: …around a factor of two…

*Done.*

Line 283: Since around 1980 CE…

*Done.*

Line 284, 285: "The WP site exhibits a seasonal distribution of precipitation typical of the Central Caucasus, with convective precipitation leading to a maximum in the summer months." Please add reference.

*Done. Reference added.*

Line 285: "The mean share of summer accumulation in the total annual accumulation was 70% (STD=18) over the course of 260 years, which is consistent with current measured precipitation data at weather stations." Since you already compiled all this data from stations etc. in the region, a Figure showing the seasonal distribution of precipitation would be very helpful (can be in the supplement).

*Done. Text revised.*
*As discussed above winter accumulation before 1865 CE (110 m w.e.) is likely underestimated due to insufficient sampling resolution. The mean share of summer accumulation in the total annual accumulation was 66% (STD=18) over the course of 144 years (1865-2009 CE), which is consistent with current measured precipitation data at weather stations. Figure S9 added.*

Line 287-290: " Over the entire period covered by the core data, there is a statistically significant decrease in the contribution of summer accumulation to the annual total (R2 = 0.6). The highest percentage of summer accumulation occurred between 1750 and 1830 CE, with an average contribution of 83% to the annual total, resulting in an anomaly of over 200% compared to the modern level in some years." Since you previously stated that the reconstructed winter accumulation is questionable in the lower (oldest) part, how sure can you be about this findings being robust or even an artefact? 1750-1830? It looks more like 1750 to around 1800 or 1810 to me. Please define "modern" (what is the reference period) and also "some years" (which years? single years or the average over the period?).

*Taken into account. Text revised. We removed this part.*

Line 290-291: "In contrast, the lowest percentage of the summer component of the annual accumulation was observed in 1935-1980 (57%)." Looking at the figures, it seems clear that for the period 1980 to the end of the record (2017?) this was even lower?

*Taken into account. Text revised.*

Line 292: "the average value of which was -25%" -25% compared to what? Which reference period?

*Removed.*

Line 315 and line 323: "However, as a correlation coefficient exceeding this value not only has statistical but also physical meaning…" and line 323: "…a physically significant radius.." I do not quite understand how the threshold value of the correlation coefficient at which a "physical meaning" starts to exists was/can be defined? Also what is/defines a physically significant radius? Please elaborate.

*Text revised.*

Line 326: "Statistical analysis of long-term series of annual precipitation (Sochi, Teberda, Krasnaya Polyana) and reconstructed accumulation on WP reveal high similarity of empirical distribution functions (Fig. 6),…" Krasnaya Polyana is not shown in Figure 6a?

*Removed.*

Line 332: "In contrast, ice-core records show a significantly larger range of variability, from a decrease of 80% to an increase of 200%." Not sure how reliable these numbers are. See previous comments regarding uncertainty of reconstruction.

*Taken into account. Rephrased.*
*Ice-core records show larger range of variability. This difference can be attributed to the length of the ice-core data, which spans 260 years, significantly longer than the longest continuous meteorological records (since 1875CE, Sochi). Consequently, the ice-core data captures the full range of precipitation variability in the Caucasus.*

Line 333 ff: "This difference can be attributed to the length of the ice-core data, which spans 260 years, significantly longer than the longest continuous meteorological records. Consequently, the ice core data captures the full range of precipitation variability in the Caucasus, ranging from extremely dry conditions (one-fifth of annual precipitation) to exceptionally wet conditions (double the annual average)." See comment above. Also, how long is the longest continuous meteorological record used here? Add in bracket.

*Done. Included.*

Figure 6: Panel b and c: The range of the y-axis has to be the same to allow for the visual comparison intended with these panels. Please adjust accordingly (e.g. -150 to 200 %)

*Done, figure adjusted.*

Line 349: Change to "…indicates that accumulation at Elbrus is primarily related to precipitation and not significantly affected by post-depositional factors (e.g. melt, wind erosion etc.). At least for temporal averages >5 years, the reconstructed record is dominated by the precipitation signal."

*Done. Text revised.*

Line 353: "However, our calendar displays considerable inter annual variability." Reference to the according table in the supplement is missing. Add "…(Table S8)".

*Done.*

Line 358: "Therefore, accounting for this factor in determining seasonal and annual precipitation amounts is critical." I do not understand why the annual precipitation amount is affected by this. The annual is the total (sum) over the entire year. Therefore, how the year is split in seasons should not matter as the sum over the full year will result exactly the same? Please comment.

*We removed "annual" for simplicity. It is actually crucial only if we are trying to compare directly with meteorological data or just to determine the season. Then annual sum can be affected too, since the annual boundary in ice core would be wrong.*

Line 359: "Failure to do so can result in errors exceeding 10% of annual precipitation sums, as illustrated in Figure 7." Annual precipitation sums, not seasonal? Why does this affect the annual sum (see above)? Also, I do not know how I can see (or derive) this 10% error from Figure 7. Would maybe including intercept and slope of the liner regressions help?

*Changed to "Figure 7 displays regression relations of the accumulation reconstructed from the core with precipitation amounts at the nearest weather stations for the 1979-2009 CE period."*

Line 350-361: "This figure displays regression relations of the accumulation reconstructed from the core with precipitation amounts at the nearest weather stations for the 1979-2009 CE period." This sentence rather seems to belong to the Figure caption of Figure 7 than in the main text.

*Done. Slightly rephrased.*

Figure 7: Please add the period of the data used (1979-2009?). Also, replace "kern" with core or ice core (in the panels, x-axis labels, itself and accordingly the figure caption.

*Done.*

Line 371, 372: "This is due to the greater spatial homogeneity of the precipitation field in the cold half of the year, which is expressed in larger values of the radii of significant correlation." This is interesting but nowhere shown. Maybe you could add a Figure, similar to Fig 5 but for summer and winter respectively to the supplement?

*Taken into account. See next comment.*

Line 373-375: "To facilitate further analysis, we have defined the cold period as the period from October to March and the warm period as the period from April to September." I like the approach to calculate and consider the seasons based on CAPE very much. It is very convincing and the improvements in doing looks very clear. Although I can understand that for reasons of simplicity this was not considered further/latter in the manuscript, I still find it a little disappointing. What might be achieved in a reasonable amount of time is to perform similar spatial correlations as performed in the following section for the ice core data (3.2.3 Comparison with gridded data). At least for one of the stations closest to the drill site (for example Terskol for which the CAPE corrected results are shown in Fig 7 and the data should thus be readily available). This would allow to immediately, i.e. visually see how the ice core reconstruction compares with the ("CAPE corrected") meteodata. Finding similar patterns as for the ice core, in my opinion, would be an additional, very strong confirmation to show that the ice core reconstructed accumulation is reflecting the regional/local precipitation signal. As a reference, to help better understand where I am going at, you might want to take a look at Figure 10 in Mariani et al., 2014. Please consider (also see next comment).

Mariani, I., Eichler, A., Jenk, T. M., Brönnimann, S., Auchmann, R., Leuenberger, M. C., and Schwikowski, M.:

Temperature and precipitation signal in two Alpine ice cores over the period 1961–2001, Clim. Past, 10, 1093–1108,

https://doi.org/10.5194/cp-10-1093-2014, 2014.

3.2.3 Comparison with gridded data (also see comment above): I am lacking a bit of insight in this section. I would assume that precipitation, also in this region is a rather local (or smaller regional) signal, especially close to an orographic barrier like the Caucasus. Therefore, a spatial correlation with a location several thousand km away as found for the summer may not be entirely convincing. The picture obtained for winter (or the pattern in S9), is what one might expected also for summer, but maybe this is really not the case in this region? I am no expert about the characteristics in this region and therefore, for readers as uneducated as myself, the suggestion made in the comment just before would be extremely helpful for further insight.

*We added another paragraph analyzing spatial correlation of the meteorological measurements at Terskol station with gridded dataset (S11). We thank reviewer for this suggestion. It certainly heled to illustrate the results and discussion.*

Line 404-417: Missing references and I also have a hard time to grasp the context to the rest of the section (see point 4 in my Main Comments).

*This section was largely rewritten; we added references and additional figure.*

*In many parts of Europe, quasi-decadal variations in atmospheric precipitation are associated with the internal nonlinear dynamics of the climate system. This is reflected in the interannual variability of indices such as NAO, AMO, EA/RW, and others in temperate latitudes (e.g., Trigo et al., 2002; Hurrell and Deser, 2009; López-Moreno et al., 2011; Ionita, 2014).*

*Negative NAO values often correspond to a weakening of cyclogenesis over Iceland and the formation of anticyclones over northern Europe, which is atypical for the North Atlantic. As a result, cyclogenesis at the polar front over the Mediterranean and Black Seas becomes more active, leading to positive precipitation anomalies in the Caucasus. Such a large-scale circulation anomaly and the negative phase of the NAO and AMO prevailed during the cold seasons of the 1960-1970 CE. The positive winter precipitation anomaly recorded in the WP ice core reflects the large-scale pattern of precipitation anomalies. In 1960-1970 CE, a pronounced region of statistically significant precipitation anomalies covered the whole of the Mediterranean and most of the southern part of Eastern Europe. The value of anomalies reached 15-20 mm per month or 20-30 % of the seasonal amount for the period October-March (Fig. 9a). They corresponded to predominantly negative geopotential anomalies in the middle troposphere, particularly over Western Europe (Fig. 9b). Northern Europe experienced an extensive precipitation deficit due to a pronounced positive geopotential anomaly over the North Atlantic.*

*The influence of the North Atlantic circulation on the cold season precipitation over the region is highlighted by the correlation of the WP winter accumulation with the Sea surface temperature and geopotential height anomalies in North Atlantic (Fig. 9a,b). For the cold period (October-March) in 1930-2009 CE, the 10-year moving averaged WP accumulation anomalies show a statistically significant negative correlation with the NAO index (r=-0.74, p<0.001) (Fig. 10c).*

Line 423: "…the 10-year moving averaged WP accumulation anomalies were statistically significantly correlated with the NAO index…" Not: ….show a statistically significant **negative correlation** with the NAO index...?

*Done. Revised.*

Line 424-425: "In the Caucasus, negative NAO values often correspond to an atypical for the North Atlantic weakening of cyclogenesis over Iceland and the formation of anticyclones over the northern Europe." Why in the Caucasus?? Should that not say: "Negative NAO values often correspond to a weakening of cyclogenesis over Iceland and the formation of anticyclones over northern Europe which is atypical for the North Atlantic." ? In the next sentence is where you then describe how this is relevant for the Caucasus region.

*Done. Revised.*

Line 435: "The substantial role of NAO in shaping the precipitation regime of southern Europe and Turkey is **was also** demonstrated in the work by (López-Moreno et al., 2011)." Please add a word or two what the cited study relies on (modelling, reanalysis data, paleoarchives,...?).
*Done. Revised.*

Line 439-443: This section is hard to read and I am not sure if I understand correctly.
What is your definition of "longer time scales" here? Maybe rephrase and restructure a bit to get the message clear? A suggestion for change could be: "Opposite to the last ~80 years where the correlation is negative, a moderate positive correlation between NAO and accumulation in the cold season was found for the earlier period from 1880-1925 (r=0.6, p < 0.001). This period is also characterized by increased summer accumulation and generally high (annual?) precipitation variability. The variation of the correlation between the reconstructed accumulation for Elbrus and the NAO index is in line with previous findings of instabilities in the connection of precipitation and large-scale atmospheric circulation at decadal timescales observed over southern and central Europe (Pauling et al., 2006)." To avoid repetition, the previous sentence referring to "longer time scales" could then be deleted.

*Done. Revised. Thank you for suggestion.*

Line 456, 457: Please add references (see comment to Line 56-59). The sentence "Thus, in the Alps, the main problem is the influence of avalanche feeding and significant blizzard redistribution of snow at the drilling sites (Bohleber, 2019)." has to be be deleted. First, the "Thus" makes no sense. Also, you cannot generally equal mountain glaciers with the Alps which is a specific mountain range in Europe.
Please note that most ice cores (also in the Alps), at least ice cores from well selected sites, are drilled

at locations where avalanches affecting the site can be excluded based on the orography. This is a generalizing but very biased statement. Again, please delete.

*Done. Removed.*

464: "Consequently, the reliability of this reconstruction is questionable." This seems to me like a subjective opinion. Please delete.

*Done. Removed.*

Line 474, 475: "The region exhibiting a stronger correlation is situated to the south of the Eastern European plain for both summer and winter seasons." This looks fairly consistent with the pattern observed for the analysis visualized in Figure 8 which might be worthwhile to point out here?

*Done. Revised.*

Line 474 ff: "However, prior to 1850 CE, the datasets do not align. The summer paleo precipitation records for the region display an unexpected and unsupported strong negative trend, with a decrease from 170 mm a-1 to 120 mm a-1."
Unexpected and unsupported? Based on what/whom? Please add a reference.

*Done. Text revise, references added.*

"Other records do not corroborate this trend" Add Reference.

*The summer paleo precipitation records for the region display a strong negative trend, with a sharp decrease from 170 mm a-1 (1750-1830CE) to ~120 mm a-1 (after 1830CE). Other records do not support such sharp change in precipitation and moisture neither in Eastern Europe (Nagavciuc et al., 2022) nor in eastern Mediterranean (Touchan et al., 2005).*

"Similarly, winter records also exhibit discrepancies before 1850, possibly attributed to a decrease in the number of observations included in the EKF400 v.2 dataset. The WP accumulation record effectively captures the decadal and long-term variability for a larger region during both summer and winter seasons."
For winter, you doubt your reconstruction for the older part (hard to find the winter layers due to resolution). Please consider the degree of uncertainty of your own reconstruction and state that some of the discrepancies observed prior to 1850 could probably as likely be due to the uncertainty of your own reconstruction.

*Similarly, winter records before 1850 also display discrepancies. These discrepancies may be attributed to a higher uncertainty in the ice core winter accumulation record, as well as a reduction in the number of observations included in the EKF400 v.2 dataset.*

Line 518: change to "…temperature-moisture regime…"?

*Done.*

**Language:**
Line 14: seasonally resolved?
*Done.*

Line 15: 181.8 m, delete hyphen between number and m
*Done.*

Line 19, 20: "Reconstructed accumulation is representative for a large region south of **the** Eastern European plain and Black sea region. Summer precipitation was found to be the primary driver of precipitation variability."
Maybe better: "Reconstructed accumulation is representative for a large region south of the Eastern European plain and Black sea region with summer precipitation being the primary driver of precipitation variability."

*Done.*

Line 26, 27: Did you mean: "Unlike for most key climate indicators, which show similar trends in most parts of the world, a strong regional variation in the sign of trend is observed for changes in precipitation amounts (IPCC, 2014)." ?
*Done.*

Line 30: In-situ precipitation measurements...
*Done.*

Line 30: … particularly for snowfall events.

*Done.*

Line 32, 33: "Currently there are 30 gridded global precipitation data sets are available including gaugebased, satellite-related, and reanalysis data sets (Sun et al., 2018)." ?
Maybe better: "Currently there are 30 gridded global precipitation data sets available, which include data based on gauge measurements, derived from satellites and reanalysis products..."

*Done.*

Line 39: ….but generally suffer from a low temporal resolution…

*Done.*

Line 41: In contrast, reconstructions from tree rings allow for annual resolution and can be calibrated...
- delete easily; this is relative and also a subjective assessment

*Done.*

Line 49: "…there is a balance between snow wind erosion and accumulation." Consider changing to: "...wind erosion and accumulation of snow is in balance... (or even better: …is close to equilibrium?)"
*Done.*

Line 67: ..an altitudinal range..

*Done.*

Fig.1 caption: (b) glaciers? I see only one glacier marked…
*Removed*

Line 96: …at the Institute of Environmental Geosciences in Grenoble (France)…
*Done.*

Line 101: … down to 168.6 m depth. …. 10 cm for the upper part…
*Done.*

Line 115: …by an increase in acidity…
*Done.*

Line 117: no **.** before Surprisingly (small)
*Done.*
Line 141: In this study, the new dating will be used as the basis for the accumulation reconstruction.
*Done.*

Line 204: …dot labels.
*Done.*

Line 207: ... from the 2009 CE drill site.
*Done.*

Line 220: …used the Global Precipitation Climatology Centre
*Done*

Line 281: The period from 1935 to 1980 CE was again characterized by relatively low accumulation in summer and relatively high accumulation in winter.
*Done*

Line 429: ….height anomalies over the North Atlantic
*Done*

Line 434: "…. (a region relatively close to the Central Caucasus) " I guess this can be deleted
*Removed.*
Line 468: … EKF400 version 2 with a 2x2 degree…
*Done.*

Line 502: …the process of layer thinning… delete the s in layers
*Done.*
Line 505: ….Energy was proposed applied, resulting… Why proposed? It was applied.
*Done.*

Line 517: This study supports….

*Done.*

---

## Author Comment (AC2)

**Reply to reviewers' comments on "History of desert dust deposition recorded in the Elbrus ice core"**

*We would like to thank both reviewers for their comments that helped us to improve and clarify the manuscript.*

Accumulation rates over past 260 years archived in Elbrus ice core, Caucasus

Mikhalenko and others.

The authors present an important new paleo reconstruction of precipitation, seasonally resolved, from the 182 m long Mt. Elbrus ice core. They apply sensible ice flow corrections to the data prior to interpretation, to allow for layer thinning, and show the regional extent of correlation between the ice core data and meteorological stations. They also demonstrate that precipitation in the region is linked to Atlantic variability, and show that the paleo reanalysis EK400v2 is shows a likely unphysical disagreement with the ice core accumulation data prior to 1850, in contrast to good correlation from 1850 to present. This should caution future users of the paleo reanalysis to view the 1750-1850 period of this product with healthy skepticism.

The paper is generally well organized and composed, although it is significantly under-referenced in many sections which is its primary weakness at present draft. The authors make many statements which need to be referenced against the relevant supporting literature. I will note some specific instances, but not all. The manuscript does need to be edited for grammar.

*We appreciate the reviewer's insightful feedback and have taken it into account during the revision process. We acknowledge that we inadvertently omitted several references in the manuscript. As per the suggestions provided by Reviewer 1, we have thoroughly revised and rewritten various sections of the manuscript, addressing the specific points raised.*

L46: this paragraph should refer to the literature of published ice core accumulation records.

*We have considered your comment regarding the absence of references in this introductory paragraph. In the subsequent paragraph, where we list the limitations and reasons that constrain accumulation reconstructions, we do cite various relevant studies to provide a more comprehensive overview.*

L50: what errors are the authors referring to? They seem to be thinking of a particular result here but do not give a reference

*Taken into account. We cite Pascal Bohleber's work here and references within since this paper investigated effects of snow scouring on isotopic records.*

*Bohleber, P., Wagenbach, D., Schöner, W. and Böhm, R.: To what extent do water isotope records from low accumulation Alpine ice cores reproduce instrumental temperature series?, Tellus, Ser. B Chem. Phys. Meteorol., 65(1), 1–17, doi:10.3402/tellusb.v65i0.20148, 2013.*

L58 "compared" to other

*Done.*

L74: how long was the 2020 core?

*Done, information added.*

Section 2.2.1: generally the author's presentation of dating and uncertainty is good, but the uncertainty of the record should be extended to graphical presentation of the data (shading, error bars, etc.).

*Taken into account, we added another paragraph considering the uncertainty. Figure 4 was updated.*

*The uncertainty range for the reconstructed accumulation values is determined by multiple factors. Dating uncertainty varies with time, providing a precision of ±1 year for the 2009-1912 CE period, ±2 years for the period 1912-1825 CE, and decreasing to ±4 years for the period 1825-1750 CE. Additionally, the uncertainty linked to the upstream effect correction is challenging to estimate precisely, depending on factors such as initial spatial accumulation distribution uncertainty, assumptions regarding its temporal persistence, and uncertainty in back trajectory modeling. To address potential errors, we estimated the uncertainty of the applied linear regression. Furthermore, for winter accumulation, the uncertainty increases due to lower sampling resolution. In particular, we added an uncertainty of 30% for winter accumulation values prior to 1865 CE, where the sampling resolution was insufficient, potentially leading to an underestimation of accumulation values.*

L271: Appreciate that the authors clearly state limitations of the data and define boundaries for interpretation

*Noted.*

L284: reference?

*Reference added.*

L359: CAPE corrections for seasonality are really well done and much needed. This is great.

*We appreciate reviewers comment.*

L384: Statement on snow deposition isn't supported by data or reference.

*Taken into account. Removed.*

L395: Section on occlusion unsupported by data or reference

*Reference added.*

L405: make sure acronyms are written out initially and defined.

*Done. East Atlantic/Western Russia teleconnection pattern added.*

695 Data availability statement isn't justified and currently doesn't align with CP policies which encourage archiving of data in agreement with FAIR principles. Authors must justify why data are only available on request, which often very must restricts access to datasets.

*We decided to publish accumulation reconstruction data in the supplementary.*

Finally, I do apologize for the lateness of this review and hope it hasn't unduly impacted the authors.